# Ultra-high dynamic range quantum measurement retaining its sensitivity

E. D. Herbschleb [1✉], H. Kato [2], T. Makino[2], S. Yamasaki[2,3] & N. Mizuochi [1✉]

Quantum sensors are highly sensitive since they capitalise on fragile quantum properties such as coherence, while enabling ultra-high spatial resolution. For sensing, the crux is to minimise the measurement uncertainty in a chosen range within a given time. However, basic quantum sensing protocols cannot simultaneously achieve both a high sensitivity and a large range. Here, we demonstrate a non-adaptive algorithm for increasing this range, in principle without limit, for alternating-current field sensing, while being able to get arbitrarily close to the best possible sensitivity. Therefore, it outperforms the standard measurement concept in both sensitivity and range. Also, we explore this algorithm thoroughly by simulation, and discuss the $T^{-2}$ scaling that this algorithm approaches in the coherent regime, as opposed to the $T^{-1/2}$ of the standard measurement. The same algorithm can be applied to any modulo-limited sensor.

[1] Institute for Chemical Research, Kyoto University, Gokasho, Uji-city, Kyoto 611-0011, Japan. [2] National Institute of Advanced Industrial Science and Technology (AIST), Tsukuba, Ibaraki 305-8568, Japan. [3] Present address: Kanazawa University, Kanazawa, Ishikawa 920-1192, Japan. ✉email: herbschleb@dia.kuicr.kyoto-u.ac.jp; mizuochi@scl.kyoto-u.ac.jp

Supreme sensitivities are realisable by exploiting the coherence of quantum sensors[1]. For quantum-sensing applications, nitrogen-vacancy (NV) centres in diamond have attracted considerable attention due to their exceptional quantum-mechanical properties[1,2], including long spin-coherence times[3,4], and due to their great potential for far-field optical nanoscopy[5–8]. Furthermore, an increase in sensitivity can be gained for alternating current (AC) field sensing by prolonging the NV spin coherence with dynamical decoupling of the centre's spin from its environment[2,3,9–12]. Therefore, AC field sensing is applied in various areas of physics, chemistry and biology: to detect single spins[13–15], for nuclear magnetic-resonance of tiny sample-volumes[16–20], for nanoscale magnetic-resonance imaging[13,21–23] and to search for new particles beyond the standard model[24,25]. In these applications, both a wide range of the AC field amplitude and a high sensitivity are very important, because the magnitude of the AC field strongly depends on the distance $r$ from the NV spin ($r^{-3}$ in case of a magnetic dipole field). This outlines the most relevant variable for this field of research: the dynamic range, which is the ratio of the range to the sensitivity, the latter being a measure for the smallest measurable field amplitude.

In previous research, NV centres were utilised for sensitive high-dynamic range direct current (DC) magnetic field measurements. A theory paper[26] discussed the application of a more general phase-estimation method[27] to a single NV nuclear spin in diamond, read out with single-shot measurements. They combined Ramsey interferometry on the nuclear spin with different delays to improve the sensitivity via Bayes' theorem applied to binary data, which precision, given full visibility, scaled as $T_{meas}^{-1}$ (with $T_{meas}$ the measurement time), dubbed Heisenberg-like scaling[28]. Adaptive[27] and non-adaptive[28,29] approaches were discussed, but they found to their surprise that under more realistic circumstances, only the non-adaptive method could still show sub-$T_{meas}^{-0.5}$ scaling, by applying different amounts of iterations in a linear way[29]. The range itself remained the same as with the standard measurement, but they improved the sensitivity for this range, hence improving the dynamic range. This theory was applied to the electron spin[30] and the nuclear spin[31] of the NV centres via the non-adaptive method. Indeed, they found that the uncertainty scaled sub-$T_{meas}^{-0.5}$ ($T_{meas}^{-0.77}$[30] and $T_{meas}^{-0.85}$[31]), while they improved the dynamic range by 8.5[30] and 7.4[31]. More recently, in an experiment at low temperature the adaptive method showed improved results, with scaling close to $T_{meas}^{-1}$ and a claimed improvement (compared to refs. [30,31]) of the dynamic range by two orders of magnitude[32].

A similar method for AC magnetic field sensing applied different order dynamic-decoupling sequences[33]. Their improvement of the dynamic range compared to a sequence with 16 π-pulses was about 26, and they explored the effect of the phase of the measured field in depth. Besides, one of the advantages of the previously reported dynamical sensitivity control[11] was the increase in the range by 4000 times, up to a theoretical maximum of 5000 times. Their uncertainty for a single measurement was about double that of a similar standard measurement, while the required multi-measurement for the large range worsened the sensitivity further (which is the uncertainty times $\sqrt{T_{meas}}$) by $\sqrt{N_\phi}$ with $N_\phi$ the number of phases applied in their method (the more phases, the larger the range, but each phase requires an additional measurement).

As to see why dynamic-range increasing algorithms are required, we look at the standard measurement. In the standard method to measure the AC magnetic field with NV centres with a synchronised Hahn-echo measurement[2,3,9,10] (Fig. 1b), after initialisation into a superposition state with a laser pulse and the first microwave (MW) π/2-pulse, the AC magnetic field is applied. Hence, the spin rotates along the z-axis, thus its phase

changes. Halfway the period of the magnetic field, a MW π-pulse flips the spin, such that the phase accumulated during the negative half of the period doubles the acquired phase. The final phase is essentially converted into a population with a final MW π/2-pulse before read-out with a laser pulse. The larger the amplitude of the field, the further the spin rotates, thus the final phase of the spin relates directly to this amplitude.

However, the phase of the spin can be determined only within 2π at best, thus the range of amplitudes is limited. If the sensor is more sensitive, the spin accumulates more phase, thus it revolves for 2π for a smaller AC field amplitude already. Therefore, the more sensitive the system, the smaller the range is. Thus, to benefit from extremely sensitive sensors which utilise entanglement[34–36] without the limitation of their minuscule range, it is important to increase this range, while retaining their high sensitivity (thus low uncertainty) as much as possible. Moreover, since the measurements of the electron spin of a single NV centre consist of iterating a sequence many times to accumulate sufficient signal (photons for NV centres), the uncertainty scales as $T_{meas}^{-0.5}$ [37].

In this work, we demonstrate and explore a non-adaptive algorithm for quantum sensors to measure AC fields with a large range for which the loss in sensitivity is negligible (thus maximising the dynamic range), both by measurement and extensive simulation. This shows that our algorithm scales nearly Heisenberg-like (here $T_{meas}^{-2}$) under realistic circumstances, thus even with the reduced contrast in the spin read-out (normally about 30% for NV centres); we explain why this happens, and its importance. Finally, we establish with our algorithm how to increase the range beyond the limit given by the best possible standard measurement, which in principle allows to extend it without bound. Throughout this paper, we use the electron spin of a single NV centre to measure magnetic fields with the phase of the spin coherence. However, the insights of this paper remain the same for similar quantum systems.

## Results

**Base algorithm**. We start with explaining the base of our algorithm (illustrated in Fig. 1), and we clarify the terms referred to throughout the paper and supplementary information. The standard measurement for AC magnetic fields, applying the Hahn-echo sequence, has a limited range $B_{range} = B_{period}/2$ due to the sinusoidal shape (with period $B_{period}$) of the signal response to magnetic field amplitudes (Fig. 1a). The sensitivity is defined as $\sigma_B \sqrt{T_{meas}}$ with $\sigma_B$ the uncertainty of the sensed quantity (here magnetic field amplitude) and $T_{meas}$ the measurement time. For this standard measurement, $\sigma_B = \sigma_S / grad_{max}$ where $\sigma_S$ is the uncertainty in the measured signal of a single measurement (in our case shot-noise limited), and $grad_{max}$ the maximum gradient in the response[3] (for example at the inflection point of the sinusoid in Fig. 1a). Therefore for these measurements, the shorter $B_{period}$ (thus the smaller the range), the steeper the slope, thus the more sensitive, as mentioned earlier.

For the maximum sensitivity, a standard Hahn-echo sequence is performed over the full period of the magnetic field (Fig. 1b, for single NV centres this period should be shorter than about half the coherence time[3]). Since the acquired phase of the spin is proportional to the area under the magnetic field curve (see Supplementary Information of ref. [3]), and hence $B_{period}$ is proportional to this area as well, by reducing the measured area $M$ times (Fig. 1b), the effective period increases by $M$ (Fig. 1a, d). The time delay between the π/2-pulses in the sequence follows from integration to compute the probed area (Fig. 1c). Hereafter, measuring an area $A$ means applying a sequence with this calculated time delay, and $A_0$ is the maximum area. Thus,

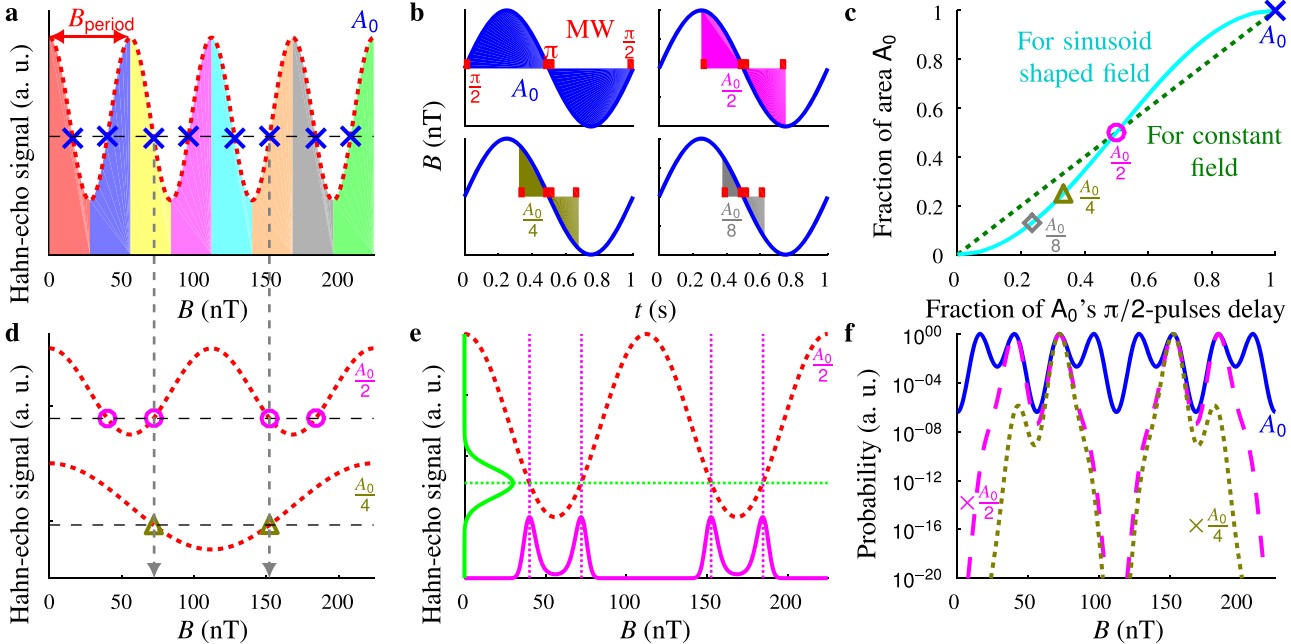

**Fig. 1 Algorithm base principle. a** Since the measured signal (red dotted line) oscillates due to the rotating spin, the magnetic field $B$ can be determined up to a certain range only. Example ranges are indicated by different colours, and the resulting magnetic field for each range given the measured signal (horizontal black dashed line) with blue crosses. **b** The conventional and most sensitive way to measure an AC field is given at the top left, with $\pi/2$-pulses of a Hahn-echo sequence at the beginning and end of the period, and the $\pi$-pulse halfway at the inflection point. The measured area $A_0$ can be reduced by moving the $\pi/2$-pulses closer to the centre (top right $A_0/2$, bottom left $A_0/4$, bottom right $A_0/8$). **c** The fraction of the maximum area $A_0$ vs the fraction of the longest time delay between the $\pi/2$-pulses for DC (green dotted line) and AC (cyan line) fields. For DC, this is linear, while for AC, this depends on the area of a sinusoid, which resembles a line near the inflection point, thus the relation becomes quadratic for short time delays. The area and thus uncertainty/range can be changed continuously by changing this delay. **d** For smaller measured areas, the probed field decreases proportionally, as does the effective frequency. In **a**, the signal for area $A_0$ was shown, while here, the signals for areas $A_0/2$ and $A_0/4$ are drawn in similar fashion, which are offset for clarity. Combining several measured areas reduces the potential fields (vertical grey dashed arrows), thus increasing the overall range. **e** After measuring the signal (horizontal green dotted line), with the known uncertainty of the signal (green line along the vertical axis), the probability distribution of the field (magenta line along the horizontal axis) follows via the sinusoidal relationship (red dotted sine-shaped line for area $A_0/2$). **f** The measurement with the largest area $A_0$ gives several similar peaks in the probability distribution (blue line). However, when combining measurements with different areas (magenta dashed line added $A_0/2$, olive dotted line added $A_0/4$ as well), the number of remaining peaks reduces, while the sharpness (thus uncertainty) remains similar to that of the first measurement.

performing a measurement with a sufficiently small area would be the simplest approach for a large-range measurement. However, roughly comparing with the measurement over the maximum area, using the same number of iterations of the sequence (thus $\sigma_S$ is similar) and the same measurement time (no optimisations), the gradient for the reduced area $\mathrm{grad}_{max,M} = \mathrm{grad}_{max,1}/M$, hence its sensitivity is $M$ times worse.

To improve the sensitivity for a large range, initially, a number of measurements with different areas are combined to uniquely define the magnetic field amplitude in a range limited by the measurement with the smallest area (Fig. 1d). Consequently, only part of the measurement time is spent on the largest area, which has the best sensitivity (but a small range), while the remainder of the time is spent on areas with a worse sensitivity. Therefore, the sensitivity of the combined measurement is strictly worse than this best sensitivity. Using halved areas (hence requiring at least $\log_2(M)$ additional areas) and the same number of iterations for each area and using no optimisations, for roughly the same $\sigma_B$, the measurement time for the combined sequence $T_{meas,M} = \lceil 1 + \log_2(M) \rceil T_{meas,1}$. Thus, the sensitivity would become $\sqrt{\lceil 1 + \log_2(M) \rceil}$ times worse, which is already a significant improvement compared to the straightforward case in the last paragraph.

The measurements resulting from different areas are combined via Bayes' theorem. For area $A_n$, the measurement gives signal $S_n$

(for example the crosses/circles/triangles on the sinusoids in Fig. 1a, d for three areas). The posterior probability distribution for the magnetic field $B$ given measured signal $S_n$ is

$$P(B|S_n) = \frac{P(S_n|B)P(B)}{P(S_n)}, \qquad (1)$$

with $P(B)$ the prior distribution, $P(S_n)$ independent of $B$, and

$$P(S_n|B) = P(S_n|S(B)), \qquad (2)$$

with $S(B)$ the relation between the signal $S$ and the applied field $B$ (the sinusoids in Fig. 1a, d, e). Fig. 1e visualises these equations. $P(S_n|S)$ is a Poisson distribution (counting photons), but it can be approximated by a normal distribution (green line along $y$-axis in Fig. 1e) when more than ~10 photons arrive (with continuity correction). This is generally the case when the uncertainty is below the maximum uncertainty, as described later. For the first measurement, the prior distribution is flat since there is no initial knowledge about the field, and for the remainder of the measurements, the previous posterior is the new prior distribution. This results in a combined distribution as demonstrated in Fig. 1f.

**Uncertainty.** Before performing measurements and simulating the algorithm, a definition of merit is required that facilitates both the sensitivity and the range. Therefore, we choose the uncertainty in magnetic field $\sigma_B$, defined as the standard deviation of

the magnetic field distribution centred around its maximum value. For sufficiently long measurement times, this gives the same result compared to applying the normal formula. However, the difference is visible for short measurement times, since it takes the range into account: we know the magnetic field is in the given range, which means that if the probability distribution is flat, the uncertainty is at its maximum $\sigma_{B,\max} = B_{\mathrm{range}}/\sqrt{12}$ (see Supplementary Note 1). $\sigma_B$ multiplied by $\sqrt{T_{\mathrm{meas}}}$ gives the sensitivity, but this is unsuitable as figure of merit at short measurement times, since its limit is $0$ nT Hz$^{-1/2}$ for $T_{\mathrm{meas}} = 0$ s while approaching the asymptotic maximum uncertainty.

At first, since the uncertainty in a range is limited by the worst uncertainty in this range, we simulated the homogeneity of the uncertainty in the complete range. For a standard measurement, the usually reported uncertainty ($\sigma_B = \sigma_S/\mathrm{grad}_{\max}$) is only true for a single magnetic field amplitude at infinite measurement time, but otherwise it is worse and inhomogeneous. By combining two measurements with the same area but their response shifted by a phase of $\pi/2$, the uncertainty becomes more homogeneous, and guarantees a lower uncertainty than the standard measurement across its range. Thus, such a measurement consists of two phases (see for example Supplementary Fig. 7a). The homogeneity is improved further by increasing the number of phases; four phases are used throughout this paper. Supplementary Note 2 describes the details of homogeneity for our algorithm, and for previous ones it is explored in ref. [38]. Since the uncertainty is nearly homogeneous, which field is applied is irrelevant while determining this uncertainty. Without prior knowledge or feedback, the uncertainty is ultimately limited by this combination of four phases for the largest area possible[3].

**Measurement compared with simulation**. For our measurements, we use an n-type diamond sample. This was epitaxially grown onto a Ib-type (111)-oriented diamond substrate by microwave plasma-assisted chemical-vapour deposition with enriched $^{12}$C (99.998%) and with a phosphorus concentration of $\sim6 \times 10^{16}$ atoms cm$^{-3}$ [3,39]. We address individual electron spins residing in NV centres with a standard in-house built confocal microscope. MW pulses are applied via a thin copper wire, while magnetic fields are induced with a coil around the sample. All experiments are conducted at room temperature. We use single NV centres with $T_2$s of about 2 ms.

We measure and simulate $\sigma_B$ for five sequences to show the consistency between the measurements and simulations, and to get an idea of the working of the base of our algorithm. Please note that the only difference between our measurements and simulations is that the simulations calculate the signal otherwise measured using the known sequence, the set magnetic field amplitude, and the parameters of the measured NV centre. The analysis applied otherwise is exactly the same, thus realistic circumstances are simulated (small contrast, shot-noise as described in Supplementary Note 2, decay due to coherence time $T_2$). The first sequence measures the largest area (here a single period of the field); the second, third and fourth use half, a quarter and an eighth of the largest area; and the fifth sequence includes these four sequences equally in a separate measurement/simulation. All include four phases as mentioned in the last subsection. Initially, the objective is to investigate the details of the algorithm itself, hence to nullify artefacts stemming from overhead times (which are implementation-dependent, and could include laser pulses, MW pulses and waiting times), these are ignored at first and explored in the discussion.

The results are shown in Fig. 2a, which reveals a number of important points. Firstly, the measurements closely match simulations. Secondly, below a certain measurement time, no

knowledge about the field is gained, and hence $\sigma_B$ is at its maximum. Thirdly, for longer measurement times, $\sigma_B$ scales as $T_{\mathrm{meas}}^{-0.5}$. Fourthly, for the combined sequence there is a region in $T_{\mathrm{meas}}$ where $\sigma_B$ scales more steeply (here referred to as the steep region). Finally, as explained in the base-algorithm subsection, the uncertainty of the combined sequence is always higher than the uncertainty of the largest-area sequence, since the former spends measurement time on sequences other than this largest-area sequence which has the lowest uncertainty. Of course, the advantage of the combined sequence over the largest-area sequence is its larger range (please remember that $B_{\mathrm{range}} \propto \sigma_{B,\max}$, see Supplementary Note 1).

**Algorithm design**. To design our eventual algorithm, its principle is explored in more detail with additional simulations. Fig. 3a shows the result for changing the relative number of iterations for each area, which reveals that there is a trade-off between the lowest uncertainty reached for measurement times at the steep region and at long measurement times. In other words, depending on $T_{\mathrm{meas}}$, a different relative number of iterations gives the lowest uncertainty. When fixing these (Figs. 2a and 3a), the uncertainty is not optimised, and thus it can display very steep curves that can be tuned to even sub-Heisenberg-like scaling (for example $T_{\mathrm{meas}}^{-4.0}$ in Fig. 3a).

For our algorithm, we optimise the relative number of iterations at each measurement time to minimise the uncertainty. The result for this measurement-time-wise optimisation is plotted in Fig. 3b. This shows that the longer $T_{\mathrm{meas}}$, the closer the sensitivity gets to its ultimate limit, where the scaling approaches $T_{\mathrm{meas}}^{-0.5}$. At the steep region of this optimum, the scaling is $T_{\mathrm{meas}}^{-0.98}$.

When we look at Fig. 3c, which depicts the relative number of iterations, we can understand how our algorithm works. For very short $T_{\mathrm{meas}}$, all measurement time is allotted to the smallest area, since the larger areas are at their maximum uncertainty and hence cannot contribute. But for longer $T_{\mathrm{meas}}$, at some time the next area becomes relevant and thus turns on, since it can receive sufficient measurement time to lower $\sigma_B$ below its maximum uncertainty. This continues until the largest area turns on, which then keeps increasing in relative importance, at which point the scaling of the uncertainty is about $T_{\mathrm{meas}}^{-0.5}$. Thus for longer $T_{\mathrm{meas}}$, the largest area receives increasingly more relative measurement time, meaning the uncertainty continuously approaches this ultimate uncertainty, as plotted by the green dashed line in Fig. 3c.

If we would increase the number of areas in the sequence, the uncertainty becomes steeper during the turning-on region (which is the steep region). Figure 3d plots the result for a large amount of areas, indicating that the uncertainty scales as $T_{\mathrm{meas}}^{-2}$ up to nearby the largest area. The scaling follows from the quadratic dependence of the area on the subsequence length (see Fig. 1c). Since closer to the largest area, this is not quadratic yet, it becomes less steep (lowest yellow crosses in Fig. 3d). The decay in coherence due to the finite $T_2$ negatively effects the uncertainty as well in this region, further decreasing the steepness. Analogue for DC measurements, the uncertainty scales as $T_{\mathrm{meas}}^{-1}$ in the steep region. Supplementary Note 3 discusses scaling in more detail beyond the indication given here. When taking any overhead time into account, the effective measurement time decreases, thus the curves would become even steeper.

So far in the examples with our algorithm, we used halved areas ($A_n = A_0/2^n$ for integer $n \geq 0$). Even though the uncertainty is mostly defined by the largest area, and the range by the smallest, the middle areas are important for reaching the lowest uncertainty (see Fig. 3c: they partake in the optimal combination). Adding more areas at integer multiples of the smallest area decreases the uncertainty, though slightly (see Supplementary Note 4).

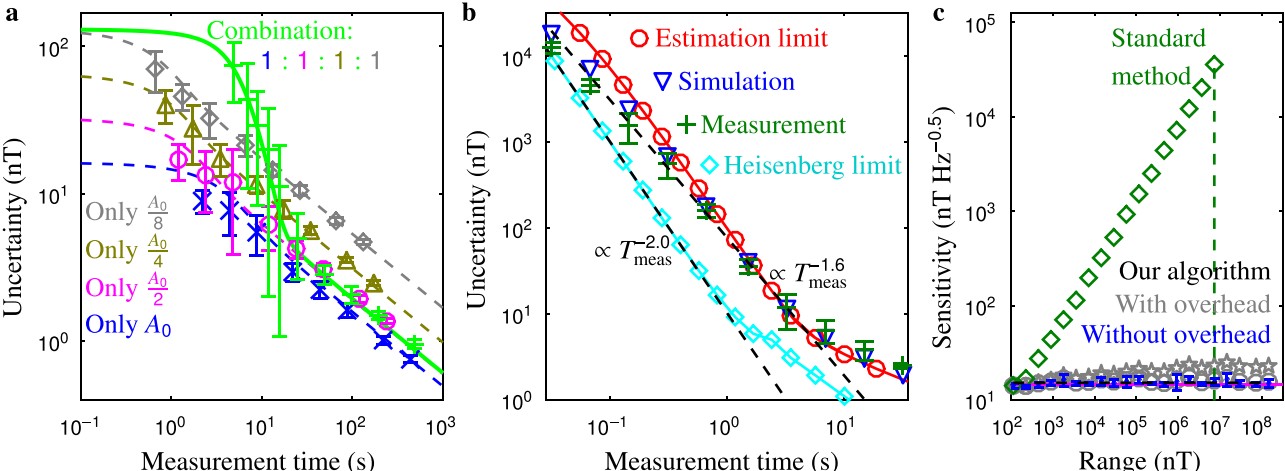

**Fig. 2 Measurement and simulation. a** Example uncertainty $\sigma_B$ vs measurement time $T_{meas}$ for AC sensing (1 kHz) comparing simulations (blue dashed line for largest area $A_0$; magenta, olive and grey dashed lines for $A_0/2$, $A_0/4$ and $A_0/8$; green line for the equally combined sequence with these areas) with measurements (blue crosses, magenta circles, olive triangles, grey diamonds and green pluses, respectively; error bars indicate single standard deviations). The simulations use the same parameters and analysis as the measurements, they are not fits. **b** Optimised uncertainty vs measurement time for AC sensing (2 kHz) around the steep region. Green pluses give the measurement results, the error bars are single standard deviations. Blue triangles are the simulation results. Red circles display the estimation of the large-range limit. Cyan diamonds plot the Heisenberg limit for infinite $T_2$ (see Supplementary Note 3 for details). The diagonal black dashed lines are guides to the eye for scaling $T_{meas}^{-2}$ and $T_{meas}^{-1.6}$. The overhead time is ignored in order to show the effect of the algorithm only. **c** Sensitivity vs magnetic field range (at 2 kHz). Blue dots with error bars (single standard deviations) give the sensitivity of our algorithm measured at each range excluding all overhead time. Grey circles plot the sensitivity including all overhead time assuming basic compact sequence design (see Supplementary Note 8), while grey pentagons plot the sensitivity assuming each area requires a separate period. The green diamonds plot the sensitivity for the standard measurement (excluding overhead time) extracted from the smallest area of our algorithm. Please note that our algorithm goes beyond the range possible with the standard measurement (vertical green dashed line) by combining non-integer-multiple areas (see Supplementary Note 6 for details). The horizontal magenta line indicates the sensitivity of the most sensitive standard measurement extracted from the largest area of our algorithm, which thus has a single small range only (the leftmost: ~$10^2$ nT). The horizontal black dashed line (on top of the magenta line) gives the fitted sensitivity of our algorithm.

**Algorithm measurement**. In Fig. 2b, measurement results of our algorithm in the steep region are plotted (for details of the measurement see Supplementary Note 5), together with the Heisenberg limit (which is only true for a small range and infinite $T_2$) and the approximate large-range limit explained in Supplementary Note 3. As mentioned before, and just like in Fig. 3d, the focus is on the scaling that originates from the algorithm, hence all overhead time is ignored. Our algorithm is very close to the limit, as could be expected since at long measurement times most time is spent on the sequence with the largest area. Moreover, our results scale approximately as $T_{meas}^{-1.6}$, which is less steep than the Heisenberg-like scaling of $T_{meas}^{-2}$, since our algorithm keeps approaching this limit.

When merely halving areas in a measurement sequence, its range is defined by the smallest area. Therefore, it would only improve the uncertainty with respect to the standard single-area measurement, but not the range. In this way, given a limit on the time delay between the $\pi/2$-pulses, for example owing to a maximum time resolution or waiting time requirements, the maximum range is restricted. However, the range of our algorithm is the inverse of the greatest common divisor of the frequencies in measured signal of all included areas (see Supplementary Note 6). For halved areas, since all larger areas are integer multiples of the smaller ones, this means that the greatest common divisor is the lowest frequency, thus the one related to the smallest area. To increase the range beyond this limit, we combine areas that are not integer multiples of each other. When purely looking at the range, combining two sequences for slightly different areas increases the range far beyond the standard measurement's range. Thus in principle, the range can be extended unlimitedly. Adding the large areas as well, it is still possible to get arbitrarily close to the ultimate uncertainty (for details see Supplementary Note 6).

The dynamic range of our algorithm is explored with measurements in Fig. 2c, which plots the sensitivity with respect to the range of the measurement sequence. Initially, for each increase in the range, an additional subsequence of half the smallest area is added. However, for the final four ranges, a single area is added at 1.5, 1.25, 1.1 or 1.05 times the smallest area. To compare with shorter sequences and with other results fairly, the sensitivity is chosen instead of the uncertainty (to calculate the dynamic range) and the overhead time is still ignored. It is computed by combining measurements from both the left side and right side of the designed range (as explained before, given the homogeneity of the uncertainty in our algorithm, the applied magnetic field does not matter). The sensitivity for the standard measurement with the same range is plotted as well (derived from the smallest area of our algorithm), and the sensitivity of the most sensitive sequence (derived from the largest area of our algorithm), the latter having a small range only (~$10^2$ nT). Our algorithm is nearly as sensitive as the most sensitive sequence, and its range can go beyond that of a standard measurement. In these measurements, the maximum range was limited by our equipment only, and could be improved further.

## Discussion

Given a fixed sequence, a subsequence contributes only to the result when the measurement time it receives is sufficiently long to lower the measured uncertainty below its maximum (see Fig. 3c). Therefore in our algorithm, the optimum sequence for a given measurement time includes contributing subsequences only. On the contrary, when combining subsequences in a fixed way with the least sensitive subsequence measured most often, for short measurement times, the more sensitive subsequences do not contribute, and hence their measurement time is wasted. This

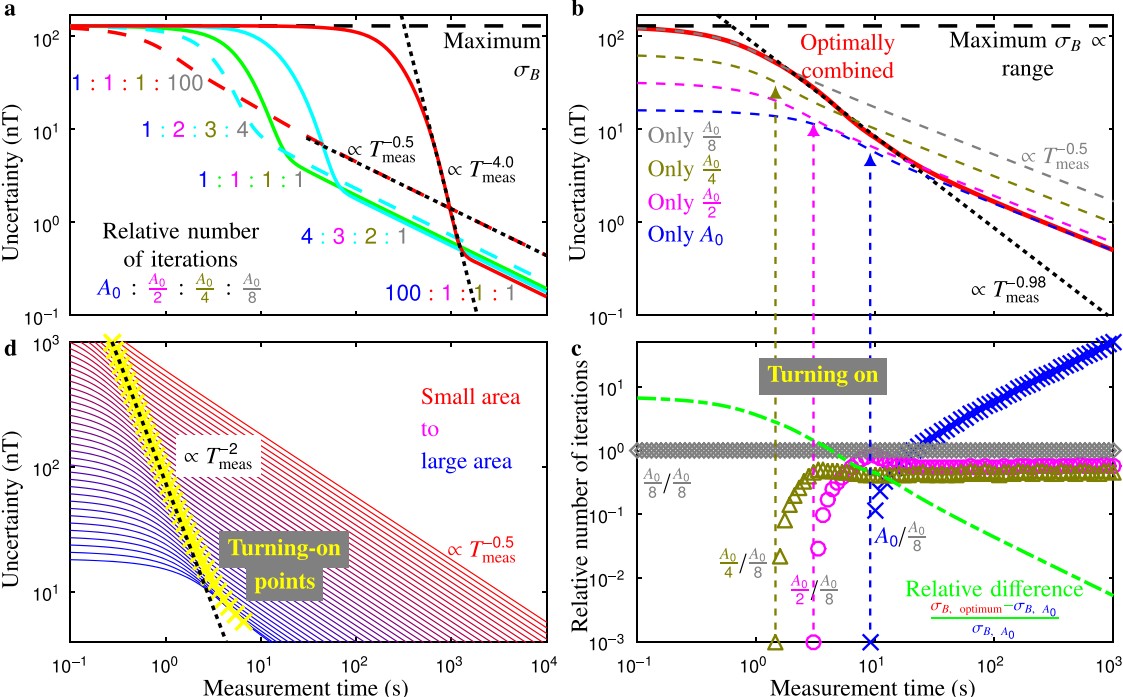

**Fig. 3 Simulation analyses. a** For different relative numbers of iterations of the subsequences (written directly left of each curve) of areas $A$, the uncertainty $\sigma_B$ with respect to measurement time $T_{meas}$ changes. Depending on the measurement time, a different combination gives the lowest uncertainty. **b** Minimised uncertainty for a large-range sequence by optimally combining the subsequences (red line). The dashed lines give the uncertainty for single-area sequences ($A_0$ blue, $A_0/2$ magenta, $A_0/4$ olive, $A_0/8$ grey). See Supplementary Note 1 for maximum uncertainty $\propto$ range. **c** The relative number of iterations for each area ($A_0$ blue crosses, $A_0/2$ magenta circles, $A_0/4$ olive triangles, $A_0/8$ grey diamonds kept at $10^0$) for each measurement time to minimise the uncertainty, which results in the red line in **b**. The vertical arrows indicate when a subsequence for its area turns on, since its relative number of iterations becomes significant. The green dashed line gives the relative difference between the most-sensitive small-range sequence (blue dashed line in **b**) compared to the optimally combined large-range sequence. This difference scales inversely with the measurement time. **d** When looking at the turning-on points (yellow crosses, fit with black dotted line) for many sequences with different areas (largest area blue line, smallest area red line), it scales as $T_{meas}^{-2}$ for short measurement times. Please note that the optimally combined result in **b** scales as $T_{meas}^{-0.98}$, since it includes relatively large areas only, equivalent to the lowest lines in this plot.

results in a steeper measurement time dependence in the same way as overhead time does. This illustrates one conclusion of Supplementary Note 3: a steeper dependence leads to a worse algorithm, since when decreasing the measurement time, the uncertainty increases more quickly for a steeper curve.

For our algorithm, if it is possible to choose for which subsequence to increase the number of iterations while measuring, the uncertainty can be minimised for all measurement times (red line in Fig. 3b), since the absolute number of iterations for each subsequence is monotonically increasing over measurement time (see Supplementary Note 7). Please note that it is known beforehand for which subsequence to increase the number of iterations, it does not depend on the measurement results, thus it is a non-adaptive method. Moreover, since our algorithm spends most time on the largest area, any overhead time (which is generally independent of the subsequence) is relatively as short as possible. This is visualised in Fig. 2c, which plots the sensitivities both with and without all potential overhead times, illustrating the overhead is negligible indeed.

For measurement times in the steep region, since quantum sensing is generally chosen for its high sensitivity, a sensor would rather unlikely be used given the high uncertainty. Therefore, this region and its scaling are fairly irrelevant: if a short measurement time is desired, less subsequences are required, which effectively puts the sensor just at the inflection point (when scaling starts to be $T_{meas}^{-0.5}$).

For measurement times beyond the steep region, $\sigma_B$ and thus sensitivity are very close to the limit for a homogeneous range.

This is still about $\sqrt{2}$ worse than the standard sensitivity for a single field at infinite measurement time (Supplementary Fig. 2). It is possible to improve towards this by applying feedback of intermediate results during the measurement, and dropping all but two phases in the process to focus on the two phases with the field to measure located at their maximum gradient, which gives the smallest uncertainty. There is a trade-off between added complexity of such an adaptive measurement[32] (real-time processing of data, changing the sequence during the measurement and/or set any phase in the measurement instead of just four with in-phase-quadrature modulation) and gained sensitivity ($\sqrt{2}$ at best for infinite measurement time), even when ignoring the processing overhead. Moreover, even under these ideal circumstances, the dynamic range, rather relevant for large-range measurements, is actually $\sqrt{2}$ worse for standard adaptive measurements compared to non-adaptive measurements (see Supplementary Note 2).

An important point ignored so far is how to implement this algorithm at all for AC fields, since its shape needs to be taken into account. As opposed to DC measurements, where the area can be reduced simply by shortening the time delay between the $\pi/2$ pulses proportionally, for AC it is more complicated, as illustrated in Fig. 1b, c. Moreover, it might seem that for each iteration another period of the magnetic field is required (resulting in the practical but non-optimal sensitivity plotted with pentagons in Fig. 2c), while for DC all measurements can be strung together, the latter limiting the measurement time. However, something similar is possible for AC fields, since the DC

part cancels, as explained in Supplementary Note 8. In our results, we neglected the effect this stringing has on the total measurement time to focus on the working of the algorithm. However, since AC stringing is only slightly less effective than DC stringing, and since often most measurement time is dedicated to the largest area (which has no stringing disadvantage), it justifies the choice to ignore the overhead time of these stringing effects. This is explored in detail in Supplementary Note 8, which describes how to design compact measurement sequences (resulting in the practical closer-to-optimal sensitivity plotted with circles in Fig. 2c). Measurements with these compact sequences illustrate that the practical sensitivity, compared to the overhead-ignored sensitivity, would worsen with about 5% when all overhead time is included using a basic sequence design (see Supplementary Fig. 10).

Additionally, please note that the description of the algorithm focussed on areas to easily translate it to any field, such as DC fields or square waves. Moreover, the frequency of the AC field is not relevant, since for lower frequencies the largest area will not span a whole period, while for high frequencies additional $\pi$-pulses are required to optimise the largest area. This defines the lowest uncertainty, which our algorithm approaches for every situation. This uncertainty increases for lower frequencies, since a smaller area is measured within the coherence-limited time delay, while for higher frequencies it decreases, due to the increase in coherence time by a dynamic-decoupling sequence (just for the larger areas, the largest defining the lowest uncertainty). Of course, the shape of the area vs time-delay graph (Fig. 1c) depends on the shape of the field and the chosen pulse sequences.

For practical implementations of the algorithm, as in the example with a single NV centre, the reader is advised that the larger the range becomes, the more prominent the effects of off-resonance MW pulses become. For DC, this is even more important (see for example ref. [31]), while for AC, the pulses are often near low fields (for example for the sensitivity-defining large area they are at the inflection points). Thus, care should be taken during the design depending on the chosen quantum system and the available technology.

As a final remark, applying the optimal number of iterations for a long measurement time gives a small chance to conclude the wrong field, since relatively little time is spent in the smaller range-defining areas (see Supplementary Note 9). However, the analysis does not return a single measured field amplitude, but a probability distribution of the field. As demonstrated in Supplementary Note 9, when the field is within a few $\sigma_B$ of the actual field, there is a single pronounced peak in this distribution. Oppositely, there are multiple strong peaks if the expected field of the measurement is significantly different. Thus, such a result could easily be discarded (of course effectively slightly reducing the sensitivity to redo the measurement for these cases).

To conclude, we have introduced an ultra-high dynamic-range algorithm for measuring magnetic fields with a quantum sensor, such as a single NV centre, for which the uncertainty, and hence sensitivity, can be arbitrarily close to the ultimate uncertainty/sensitivity by increasing the measurement time. The maximum range depends on the smallest difference in areas attainable, which results in a larger range than possible with a standard measurement. As example, we demonstrated a dynamic range of $\sim 10^7$, an improvement of two orders of magnitude compared to previous algorithms[32] (please note that a fair comparison between algorithms corrects the results for the coherence time, the minimum/maximum time delays, the applied spin-measurement method and the experimental equipment, as these change the results independent of the applied algorithm). Moreover, we explained the origin of Heisenberg-like scaling in algorithms and why steeper scaling indicates a worse algorithm. Our algorithm

and its implications are the same for other modulo-limited sensors, thus it paves the way to optimally benefit from extremely sensitive entanglement-based sensors for large-range applications.

## Data availability

The data that support the findings of this study are available from the corresponding author upon reasonable request.

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

## Acknowledgements
The authors acknowledge the financial support from MEXT Q-LEAP (No. JPMXS0118067395), KAKENHI (No. 15H05868, 16H06326) and the Collaborative Research Program of ICR, Kyoto University (2019-103). They also thank Prof H. Kosaka for helpful discussions.

## Author contributions
E.D.H. designed the algorithm, performed the experiments/simulations/analyses and conceived the supplementary; H.K. grew the phosphorus-doped diamond, assisted by T.M. and S.Y.; N.M. supervised the work; E.D.H. and N.M. wrote the manuscript, and all authors discussed it.

## Competing interests
The authors declare no competing interests.

## Additional information

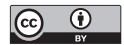

