## [Peer Review File · Nature Communications]

Reviewers' Comments:

Reviewer #1:

Remarks to the Author:

Authors implement the so-called non-adaptive phase estimation algorithm (NAPEA) on NV spin qubit for sensing AC magnetic fields. Their implementation involves the use of different pulse-echo timings to enable the different phase accumulations needed for NAPEA. I find this interesting and the manuscript is well organized, thus would like to see this published in Nat Commun provided that authors address the following concerns:

1. Authors make good reference to NAPEA implementation for DC fields but missed one important and relevant work wherein NAPEA was adapted for AC fields:

DOI:10.1103/PhysRevB.88.220410

In the above work, in order to get the different phase accumulations needed for NAPEA, authors used different order dynamic decoupling whereas in this work the order stays same but uses different pulse pulse intervals.

2. Calling this "unlimited dynamic range" sounds overambitious and not entirely accurate. There's always a limit constrained from finite pulse-lengths. Attempting to shorten the pulse-lengths requires having high Rabi frequency but at some point the rotating-wave-approximation collapses. So I would suggest authors use a more reasonable term such as "high dynamic range" instead of "unlimited...".

3. In 3-30: "By combining two measurements... the uncertainty becomes homogenous.." is not entirely accurate. Better replace it by "the uncertainty becomes less inhomogeneous.."

Also, authors claim in 4-5 "This can be improved by increasing the number of phases;.." There is another previous work authors should be citing here:

DOI:10.1103/PhysRevB.90.024422

The above computational work presents how the consistency in sensitivity improves with introducing more phases. Given there is only few work on PEA, it is unfortunate authors miss these relevant publications.

4. Noted multiple places where the text don't read well:

1-70: The sentence "To fully exploit..." is confusing.

2-35: "also to clarify" perhaps should be replaced by "and clarify"

2-60: "referred to via"

2-70: ".. different areas is combined.." should be "..are combined.."

2-80: " The sensitivity of the combined measurement.." Hard to follow. Break into two?

Reviewer #2:

Remarks to the Author:

This paper presents an algorithm to combine Hahn echo sequences of various delay times to measure the amplitude a AC field with known frequency. The algorithm is claimed to be able to achieve a dynamic range up to the inverse of the time resolution of the sequence yet with a good sensitivity. The claimed features of the algorithm are of potential importance in parameter estimation of AC fields. However, the writing of the paper and the SI is very unclear, which makes

it hard to judge whether the claims are well justified. I spent a lot of time reading the paper and the SI many times and could still only get some vague ideas of what is done in the paper. I cannot recommend publication of the paper before it is rewritten to be clear enough.

Below are a few specific points that need the authors to clarify (I may have more questions and more relevant questions when the paper is rewritten clearer):

1. The English of the main text and the SI needs thorough rewriting. Many sentences do not even have correct grammar. Many more are unclear.
2. The frequency of the AC field is not given. Since T_2 (2 ms), which limits the coherent measurement time, is given, the absolute value of the frequency of the AC field is relevant (it determines the longest delay time in the Hahn echo pulse sequences).
3. T_{meas} has never been defined in the paper. One can only guess. And it might have been used with different definitions throughout the paper. Is it the duration of the Hahn echo sequence (T_h)? Is it the Hahn echo sequence times the number of phases ($T_A = T_h \times \text{\#phases}$)? Is it the duration of the Hahn echo times the number of phases times the number of iterations ($T_{\text{sequence}} = T_A \times \text{\# repetitions}$)?
4. How are the measurement sequences arranged? For example, consider a sequence of $A_0/8$ during $(T/2 - \tau, T/2 + \tau)$ (where T is the period of the AC field, τ is the delay time of the Hahn echo sequence), is the system idle during the waiting periods $(0, T/2 - \tau)$ and $(T/2 + \tau, T)$? Are these waiting periods counted to T_{meas} ? If not, the definition of the sensitivity is wrong.
5. The different phases of the measurement should be explained. Not all readers are familiar with such technical terms.
6. When there are different A_n , different phases, and different repetitions, how are measurement sequences arranged? How is T_{meas} defined?
7. What is the value of σ_S used in the simulation? How is such a value determined?
8. Why is the algorithm worse when the sensitivity curve is steeper? Should it be the opposite?
9. In Figure 3, the optimization is for what? It appears the optimized curve has uncertainty worse than the A_0 curve (Fig. 3b).
10. In Fig. 3b, how could the relative difference be less than 1 at the long time limit (since in Fig. 2b the optimized uncertainty is almost one order of magnitude larger than the A_0 curve)?
11. In the figures (both in the main text and in the SI), too many curves and symbols are plotted in similar colors, similar shaped, overlapped, crowded, etc, very difficult to be distinguished by naked eyes.
12. Page 2, line 81: Why is $T_{\text{meas},M} = [1 + \log_2(M)] T_{\text{meas},0}$? If $M=1$, would $T_{\text{meas},1} = T_{\text{meas},0}$? Again, what is the definition of $T_{\text{meas},M}$?
13. The experimental method is not described.

Reviewer #3:

Remarks to the Author:

The authors devise a non-adaptive algorithm for high dynamic-range measurement of modulo-limited sensors. Principally, it can realize arbitrary dynamic-range measurement with retaining the highest sensitivity, which is practical for realistic applications. The algorithm is elaborated and the corresponding experiments and simulation are well done. However, the core idea of the algorithm does not have much novelty and has been used in common experiments by intuition. It can be illustrated simply: when the most sensitive way using coherence time adequately measures a large magnetic field beyond its dynamic range, it will lead to the ambiguity of multi-peak as shown in Fig. 1f; but it can be simply remedied by adding the experiments of using a shorter interference time but having a larger dynamic range to kick out wrong peaks; the process of kicking out is effective because it can use something like dichotomy resulting in a logarithmic time overhead; moreover, these remedying experiments do not require high precision, thus their duty cycle can be small enough at infinite measurement time. Since the journal of Nature Communication requires papers with enough novelty, I do not recommend this paper.

Questions on some details of the paper:

1. What combination of iteration numbers of each subsequence is used for blue crosses and red circles in Fig. 2b? It is different from the equally combined sequence in Fig. 2a.

2. Why the uncertainties of the measurements with a single area are below than combined one as shown in Fig. 2a and Fig. 3b? Notice that the choice of a single area is also a special combination. I guess that the initial set uncertainties are different and within dynamic ranges respectively. It may cause confusion and need some explanation.

3. In the row 38, is the formula $\text{Brange} = 1/(2B\text{period})$ typed incorrectly? Should it be $\text{Brange} = B\text{period}/2$?

Reviewer #1

Reviewer's comments: *Authors implement the so-called non-adaptive phase estimation algorithm (NAPEA) on NV spin qubit for sensing AC magnetic fields. Their implementation involves the use of different pulse-echo timings to enable the different phase accumulations needed for NAPEA. I find this interesting and the manuscript is well organized, thus would like to see this published in Nat Commun provided that authors address the following concerns:*

Our reply: We thank the reviewer for the time to read and comment on our manuscript. We do want to comment on the description given above. Although our implementation is indeed a non-adaptive algorithm, we do not think there is a "the" NAPEA. There are many different ways to implement a non-adaptive algorithm, such as given in Ref. 30, and another way is given in the suggested citation below (now Ref. 37). Neither any of these previous papers, nor our manuscript, makes any claim on having reinvented the mathematics or principles generally used for such algorithms. The main novel part of our manuscript is not the application of a NAPEA on AC, the choice for AC was made to enable a more thorough explanation of all details, it could as well have measured DC or any other field shape. The first novel part, relevant here, is how the sequence is built. As opposed to previous attempts, this allows for approaching the limit of the sensitivity (given the coherence time). This novel part is therefore expressed in the second part of the title: "retaining high sensitivity": there is no practical loss in sensitivity while the range is increased. This was not the case for the previous approaches, where linear combinations were used.

A second important feature stemming from our algorithm design is the manifestation of Heisenberg-like scaling. In previous works, Heisenberg-like scaling was observed, but this can be partly contributed to an artefact. For example, in Refs. 30, 34 and 35, the number of iterations of each subsequence is increased according to a linear formula with most iterations for the sequence with the shortest delay (the algorithm in these references are explained via delays, which is a clear description for DC when no constraints, such as mentioned by the reviewer in comment 2 below, are present). Thus, what roughly happens is that for short measurement times, the subsequences with long delays do not get sufficient measurement time to get beyond their flat regions. Therefore, they do not contribute to decreasing the uncertainty of the result, only the subsequence with the shortest time delay does. This means that part of the measurement time is effectively wasted, and part (the part of the shortest time delay) estimates the field/phase. When increasing the measurement time, at some point the next subsequence (one-before-shortest delay) gets beyond its flat region, and it will contribute. Therefore, there are two ways in which the steep scaling is obtained: the first because of the Heisenberg effect since a subsequence with a longer delay starts to contribute, the second because initially most of the measurement time was not contributing to the result, thus the effective measurement time was short (for example 10% with 1 s out of 10 s), but now the effective measurement time increased (for example 20% with 4 s out of 20 s), which thus also makes the curve steeper. In our algorithm, only the first, thus the Heisenberg effect, is contributing, since no measurement time is spent on any subsequence that cannot contribute yet (this is shortly discussed in the paragraph starting at page 7 line 4).

Below we will address the comments of the reviewer, starting with comment 2, since this allows for continuation of the above.

Reviewer's comments: *2. Calling this "unlimited dynamic range" sounds overambitious and not entirely accurate. There's always a limit constrained from finite pulse-lengths. Attempting to shorten the pulse-lengths requires having high Rabi frequency but at some point the rotating-wave-approximation collapses. So I would suggest authors use a more reasonable term such as "high dynamic range" instead of "unlimited..."*

Our reply: This question brings us to the second novel point of our algorithm. Before discussing this, we would like to give a thought on the example given regarding the limit constrained from finite pulse-lengths. Even when the rotating-wave approximation breaks down due to the Bloch-Siegert shift, this does not affect our algorithm, merely the analysis. Suppose the pulse lengths are made very short, then the read-out result vs magnetic field strength is not a sinusoid anymore, probably it is even not a periodic function anymore but more like a function that oscillates with a frequency that slightly changes over time. Neither do effect the algorithm though, just the analysis. This is actually also mentioned in the second citation suggested by the reviewer (page 3, first paragraph of current Ref. 38). As mentioned in Eq. (2) in our manuscript (page 2 line 103), the relation between the field and the read-out signal is simply a function. If it is a sinusoid, the analysis is relatively simple. If not, then the shape has to be taken into account and for example some calibration measurements (or computations) will be required to set up the measurement. However, this is not a limitation of the algorithm. Nonetheless, we reckon that, although with the explanation with delays it is not as clear as in our algorithm, the previous algorithms would probably be able to make similar adjustments. Moreover, it is more of an “experimental” constraint than an algorithmic constraint. If the clearer adjustment to finite pulse widths would be the only point of our algorithm, we would have used the same as used in previous papers: “high dynamic range”. However, it is not.

This brings us to the important difference. In previous works, there was no increase in range compared to the standard measurement, only the sensitivity with respect to the standard measurement was improved. Thus, since the maximum range therefore depends on, as mentioned by the reviewer, finite pulse lengths (for the sake of discussion ignoring the previous paragraph), the range is high but limited. In our case, we go beyond the range of the standard measurement, as explained in the results section (paragraph starting at page 5 line 101, and mentioned at various places such as in the abstract page 1 line 12 and in the introduction page 2 line 33), shown in Fig. 2c and explained in more detail in Supplementary Note 6. Just focussing on the range, in principle two areas of slight difference would increase the range far beyond the standard measurement one. Since the pulse widths and pulse distances will still be relatively large, there is no constraint from these. Hence, in theory the range is unlimited. We guess the constraint would be of technical nature: how accurately can the pulses be timed. However, even applying just 100 T in an experimental setup, which is still a small field for this algorithm, will prove challenging, thus it is hard to define a concrete limit. Please note that we used this principle in the dynamic-range measurement (Fig. 2c). We have made it clearer in the description of the experiment (page 6, line 15).

Since in the first part of the title we want to express this novelty of our algorithm, we chose “Unlimited dynamic range”. “High dynamic range” would not cover the contents, as it is the same as previous titles. Given that the constraints are of a technical nature that would not limit the range for practical applications, thus making it practically unlimited as well, we feel that our choice was fair. Given the above description, we now changed it to “Practically and theoretically unlimited”, but we do think that without the adverbs the title is clearer.

Reviewer’s comments: 1. *Authors make good reference to NAPEA implementation for DC fields but missed one important and relevant work wherein NAPEA was adapted for AC fields:*

DOI:10.1103/PhysRevB.88.220410

In the above work, in order to get the different phase accumulations needed for NAPEA, authors used different order dynamic decoupling whereas in this work the order stays same but uses different pulse-pulse intervals.

Our reply: We thank the reviewer for suggesting a paper about a different implementation of a non-adaptive algorithm, which we have added in the introduction (page 2, line 7; Ref. 37). We're afraid that the title of the above paper made it hard to find via literature research (we focused on dynamic-range papers and their predecessors, such as earlier papers about PEA), but it is a very relevant paper indeed.

Reviewer's comments: 3. In 3-30: "*By combining two measurements... the uncertainty becomes homogenous..*" is not entirely accurate. Better replace it by "*the uncertainty becomes less inhomogeneous..*"

Our reply: We had the theoretical case from Supplementary Note 2 in mind, but indeed this is not described like this in the main text. We have changed it (to "becomes more homogeneous", page 4 line 4).

Reviewer's comments: Also, authors claim in 4-5 "*This can be improved by increasing the number of phases;..*" There is another previous work authors should be citing here:

DOI:10.1103/PhysRevB.90.024422

The above computational work presents how the consistency in sensitivity improves with introducing more phases. Given there is only few work on PEA, it is unfortunate authors miss these relevant publications.

Our reply: The suggested citation is indeed a very nice work which would have been useful to know beforehand, making the explanation about phases easier for the readers. Now, we refer to this work explicitly (page 4, line 12). Our approach in the supplementary is different though (but the conclusion with respect to phases is of course the same) and makes a number of different points relevant to our manuscript as well, so we feel it still contributes to this paper. Now, we also refer to the above paper in the supplementary notes to allow the audience to learn more about phases (last paragraph dedicated to it: supplementary page 5). We hope that the reviewer does understand that this is indeed just unfortunate, this paper never popped up, since it postdates the referenced earlier papers about high-dynamic range including the one of the same authors (Ref. 34), while the later paper (Ref. 36) does not refer to the above suggestions, thus making it hard to find.

Moreover, after reading this paper, since the reviewer suggests this is one of the few papers about the topic, we have adjusted our manuscript to be more consistent with terminology. This is especially true for the meaning of dynamic range, which was used interchangeably with the meaning of just range. Now, we define dynamic range in the introduction (page 1, line 41), and we only use "dynamic range" when specifically meaning this, while otherwise we use "range". Please note that, since the definition of "dynamic range" is not consistent in previous papers, we chose for the ratio of maximum range to sensitivity (most similar to the above citation, and Ref. 34), since this ratio is often given in previous papers as the sensitivity allows for fair comparison. For example, a definition ignoring time (like range over smallest measurable field) would allow to extend the dynamic range to infinity just by measuring for an infinite time.

(Just to be sure, as a side note, since this is probably just a matter of definition, we would like to mention that we do not consider the algorithm in our manuscript a phase-estimation algorithm. A phase-change-estimation algorithm would be more appropriate, but still not correct. Effectively, it is an estimation algorithm of the change in phase that would have occurred if we would have held the quantum system in a certain field while using a time delay optimising the sensitivity.)

Reviewer's comments: 4. *Noted multiple places where the text don't read well:*

1-70: *The sentence "To fully exploit..." is confusing.*

2-35: *"also to clarify" perhaps should be replaced by "and clarify"*

2-60: *"referred to via"*

2-70: *".. different areas is combined.." should be "..are combined.."*

2-80: *" The sensitivity of the combined measurement.." Hard to follow. Break into two?*

Our reply: We thank the reviewer for helping to clarify the manuscript, and we rewrote all of the above suggestions.

1-70: rewrote the whole sentence and relocated it to a clearer place (page 1, line 68).

2-35: changed to: "and we clarify" (page 2, line 40).

2-60: rewrote the whole sentence (page 2, line 68).

2-70: "a number of" is plural indeed (page 2, line 79).

2-80: rewrote the whole sentence, resulting in two sentences (page 2, line 82). Moreover, since the other reviewers had questions related to this topic as well, we added a similar explanation when the first measurements are discussed, since the measurement results illustrate it well (page 5, line 15).

Reviewer #2

Reviewer's comments: *This paper presents an algorithm to combine Hahn-echo sequences of various delay times to measure the amplitude of an AC field with known frequency. The algorithm is claimed to be able to achieve a dynamic range up to the inverse of the time resolution of the sequence yet with a good sensitivity. The claimed features of the algorithm are of potential importance in parameter estimation of AC fields. However, the writing of the paper and the SI is very unclear, which makes it hard to judge whether the claims are well justified. I spent a lot of time reading the paper and the SI many times and could still only get some vague ideas of what is done in the paper. I cannot recommend publication of the paper before it is rewritten to be clear enough.*

Our reply: We appreciate that the reviewer took the time to read and comment on our manuscript. Sadly enough, in its current condition, we were not able to clearly transmit our message to the reviewer. Of course, we want to reach out to the whole diverse audience, so we will try our best to clarify all points as much as possible.

For starters, we would like to comment on the initial remarks. The point of the algorithm is not to use Hahn-echo sequences, but to measure different areas. For AC, this could be done with Hahn-echo sequences, it would be FID sequences for DC, etc. Since we chose to explain the algorithm with AC, since this will elucidate the features better, we happen to have used Hahn-echo sequences. We will address this further in the reviewer's comment 2 below.

Next, there is no claim in the manuscript that the algorithm will "achieve a dynamic range up to the inverse of the time resolution of the sequence". The claim is unlimited. But to put this into words, the claim is the inverse of the smallest area difference attainable, with for DC without extra measures might be indeed related to the time resolution (but it is not its inverse, but related to the area measured during this time), while for AC the non-linearity of the field places it beyond this. In our case, we go beyond the range of the standard measurement, as we explained in the results section (paragraph starting at page 5 line 101), shown in Fig. 2c and explained in more detail in Supplementary Note 6. If we just focus on the range, two areas that are slightly different would increase the range far beyond the standard measurement one. To clarify this point, we have left the practical details to the SI (SI 6), and focused on the principle in the main text (page 6 line 4). Moreover, the meaning of "dynamic range" was not made clear in the initial manuscript and it was used interchangeably with "range", which could lead to confusion. Now, it is defined in the introduction (page 1, line 41), and the terms "dynamic range" and "range" are used correctly in the manuscript and the SI.

Also, our claim is not to get just a good sensitivity, our claim is that the sensitivity is practically the same as a standard minuscule-range measurement. So high reward, negligible cost. This is one of the main messages of our manuscript, and it is mentioned in the abstract, in algorithm design, and in the conclusion. We hope that the changes with respect to the comments below made this more visible.

All claims are justified by explanation, simulation and measurement throughout the manuscript and SI. We hope that with the adjustments described below this will become clearer.

Reviewer's comments: *Below are a few specific points that need the authors to clarify (I may have more questions and more relevant questions when the paper is rewritten clearer):*

Our reply: We addressed them point by point.

Reviewer's comments: 1. *The English of the main text and the SI needs thorough rewriting. Many sentences do not even have correct grammar. Many more are unclear.*

Our reply: It would have been very helpful for us to write down the line numbers for each page with errors, as reviewer #1 did. Even after another proofread by an external native speaker, no additional errors were found (so apart from the one given by reviewer #1), but we did change the clarity of some sentences (word order/choice of words). We take great care in preparing our manuscript, so we do want to know if there are any mistakes in English. Please do note that “staccato language” is used in for example figure captions and explanatory comments (between brackets) to remain concise, as is commonly done (but its usage probably depends on personal preference).

Regarding to the “many more are unclear”, in order to improve the quality, please do let us know which sentences the reviewer considers unclear. See reviewer #1 comment 4 for an example.

For now, we assumed that the reviewer's “many unclear sentences” are the same as the ones given by reviewer #1, in combination with the comments below, and we have made improvements for all.

Reviewer's comments: 2. *The frequency of the AC field is not given. Since T_2 (2 ms), which limits the coherent measurement time, is given, the absolute value of the frequency of the AC field is relevant (it determines the longest delay time in the Hahn echo pulse sequences).*

Our reply: The changes we made are described in the next paragraph, but initially we would like to explain why the frequency is not that relevant in our algorithm: the frequency does not determine the longest time delay in the sequence, since the Hahn echo sequence is merely an example. Thus, T_2 determines the best uncertainty, but it does not relate to the frequency. For clarity, the confusion might stem from that for the standard measurement (for AC the Hahn-echo measurement) it does, but not for our algorithm. In the example, for simplifying the explanation, the largest area is chosen as the standard Hahn-echo measurement, which is a relatable measurement for many researchers (now indicated at page 5 line 1). Then, the areas are decreased in size, eventually measuring the area of a straight line. However, the largest area can also start at the linear region (for a lower frequency). For much higher frequencies, the initial area should not be measured with a Hahn-echo sequence, but with a dynamic-decoupling sequence. To minimise measurement time, even asymmetric Hahn-echo sequences could be used. In other words: the important point is choosing which areas to measure; how to measure these areas depends on the signal: the shape, for AC the frequency, and so on.

In order to clarify this, we have added a paragraph in the discussion (page 7, line 88). Moreover for completeness, we have added the chosen frequencies for the measurements in Fig. 2 in its caption. Please note that although we did not choose to use the optimal frequency (about 1 kHz) for the final measurements (to get the dynamic range), meaning the sensitivity could be lower, this does not affect the conclusion. When comparing algorithms (as mentioned on page 8, line 18), only algorithmic details should be taken into account, while others (such as our superior coherence time) should not. Thus using 1 kHz would improve the sensitivity by very roughly $\sqrt{2}$, but in the comparison we would remove this $\sqrt{2}$, leading to the same improvement over the previous algorithm.

Reviewer's comments: 3. *T_{meas} has never been defined in the paper. One can only guess. And it might have been used with different definitions throughout the paper. Is it the duration of the Hahn echo sequence (T_h)? Is it the Hahn echo sequence times the number of phases ($T_A = T_h \times \text{\#phases}$) ?*

Is it the duration of the Hahn echo times the number of phases times the number of iterations ($T_{\text{sequence}} = T_A \times \# \text{ repetitions}$)?

Our reply: Since we are not sure what exactly is meant with definition (as in, definition of the symbol T_{meas} or of the meaning), just in case we would like to mention that at the first occurrence of this symbol, (page 1 line 68 in the original manuscript, now page 1 line 76), it is defined. Moreover, on the first occurrence in the results section (page 2 line 42 in the original manuscript, now page 2 line 48) it is defined a second time. In the SI it is also defined on its first occurrence. To improve clarity further, now, we define it once again on the first occurrence in a figure (page 4 first line of the caption).

For the second option, the definition as such does not really change (it is the time it takes to do the measurement, so of course all phases and iterations are included), but what is included with respect to overhead time depends on the specific measurement and its goal. This is important for understanding the results, and we give more details about this in related comments 4 and 6 below. We do think that it is unreasonable to think that any part of the measurement sequence that measures the field would be ignored (such as phases, areas, repetitions), so we do not specifically mention that they are included. We hope that the changes with respect to overhead time described in comment 6 will be sufficient for the reader to understand the contents of the measurement time.

Reviewer's comments: 4. *How are the measurement sequences arranged? For example, consider a sequence of $A_0/8$ during $(T/2 - \tau, T/2 + \tau)$ (where T is the period of the AC field, τ is the delay time of the Hahn echo sequence), is the system idle during the waiting periods $(0, T/2 - \tau)$ and $(T/2 + \tau, T)$? Are these waiting periods counted to T_{meas} ? If not, the definition of the sensitivity is wrong.*

Our reply: The answer to this question is given in the discussion (paragraph at page 7 line 65 is dedicated to it) and further explored in SI 8. Initially, overhead time is neglected (please see comment 6 below for details per measurement and how we addressed this). In the discussion, this is justified, and a visual explanation is given in SI 8. It explains that there is no need to measure around $T/2$ (so there is little idle time away from the inflection point), and thus such overhead time is not significant. Overhead time related to laser pulses becomes the most significant source of overhead for large ranges, which is the same for DC. Moreover, when approaching the lowest uncertainty, all of these overhead effects become very small since most time is spent on the largest area, which has no additional stringing overhead (page 7 line 81). We will explain all details in comment 6 below.

Reviewer's comments: 5. *The different phases of the measurement should be explained. Not all readers are familiar with such technical terms.*

Our reply: We indeed only implicitly introduced phases. We have added the explanation explicitly at the first mentioning of phases by changing its introduction (page 4 line 4) and by adding a sentence (page 4 line 6), and now we refer to SI Fig. 7a which happens to show two phases which visually explains it to the reader.

Reviewer's comments: 6. *When there are different A_n , different phases, and different repetitions, how are measurement sequences arranged? How is T_{meas} defined?*

Our reply: This is an important question, which was partly left to the discussion and SI. In the next paragraphs, we will explain the meaning of T_{meas} for each measurement. As a general comment,

for this manuscript, the focus is on explaining the algorithm. Thus, for clarity, several complexities that are not that relevant for understanding or explaining the algorithm are left for the discussion. One important point was discussed in comment 4 above. As will be clear from comment 4, even when taking overhead into account, the conclusions will only slightly change quantitatively, but not qualitatively (see below as well). With respect to the measurements, since the order etc do not matter for the explanations, the sequences are put in a single long sequence from the largest to the smallest area. In principle, if this is not possible for other researchers due to technical limitations, a sequence could be split up and measured sequentially in any order, nothing effects the qualitative results.

To clarify the choices for measurement time for each measurement, we have made the following changes. For the measurements of Fig. 2a, overhead time of pulses etc are taken into account, but stringing effects are not. The former is mentioned in the text (page 4 line 40), but not specifically in the figure caption, since the point of Fig. 2a is merely explanatory.

For the measurements of Fig. 2b, all overhead time is ignored. The reason is, just like in Fig. 3d, the focus is on the effect of the algorithm itself on scaling. Any overhead time would increase the steepness and thus obscure the effect imposed by just the algorithm (mentioned at page 5 line 75, and in the discussion at page 7 line 14). For Fig. 3c this was already mentioned in the caption (page 6 caption 3d) and in the text (page 5 line 64). For Fig. 2b it was already mentioned in the caption, and we have added it to the text about the measurement (page 5 line 92).

For the measurements of Fig. 2c, all overhead time is ignored. The reason is that this is the only way to allow fair comparison with other results. On the one hand, for comparison with the standard measurement, since overhead has a very significant effect on the standard measurement, thus including overhead would make our algorithm look much better. On the other hand, for comparison with other results. The previous best algorithm, to which we compare in the conclusion (page 8 line 18), also ignores any overhead. Please note that this is commonly done, since the assumption is that in the future, measurement methods might improve, thus improving results of any algorithm, thus the desire is to focus on the effect of the algorithm alone. Again, we want to fairly compare results, and since the overhead time of our algorithm is relatively small (page 7 line 29), even when including all effects of any overhead, our sensitivity would increase a few percent only (now added in Fig. 2c and explained with an additional SI Fig. 10 and an additional paragraph in the SI page 13 at the bottom and mentioned in the main text at page 7 line 83). However, the sensitivity of the mentioned previous best algorithm (page 8 line 18) would worsen by almost an order of magnitude, meaning we could say our algorithm is actually three orders of magnitude better instead of two. We feel that it is unfair to compare the algorithms based on non-algorithmic parameters, and thus we chose to ignore all overhead time as is normal, and we kept the mere two orders of magnitude. Now, we have added this to the caption of Fig. 2c and to the text (page 6 line 20). And as mentioned above, we updated SI 8 with another figure and its explanation, and we refer to this in the discussion (page 7 line 83).

Reviewer's comments: 7. *What is the value of σ_S used in the simulation? How is such a value determined?*

Our reply: The value is based on shot-noise, which is the limiting noise for the quantum sensor used in our experiments (a single NV centre). This is fairly basic and described in many papers (as referred, most conveniently in Ref. 3). Here, it is simply $1/\sqrt{N N_{ph}}$, with N the number of iterations of a sequence and N_{ph} the number of photons received for each read-out pulse in the sequence (only counting the photons that are used to distinguish between the spin states). The latter, indeed a parameter for the simulations, follows directly from the experiments (repeat N times, count the photons), and this shows that the formula is consistent (as expected for shot-noise limited

experiments). For the audience outside this field of science, one point of Fig. 2a is to demonstrate that indeed, as expected, the simulation using this σ_S and the measurement being affected by shot-noise lead to the same results.

Nonetheless, we did not specifically mention this in the manuscript. We have added a reference to the SI when starting to discuss simulations while mentioning shot noise (page 4 line 41), and in this SI we added a paragraph to introduce the formula (SI 2 paragraph starting at page 3 line 32).

Reviewer's comments: 8. *Why is the algorithm worse when the sensitivity curve is steeper? Should it be the opposite?*

Our reply: The confusion stems from mixing two different scenarios, as discussed in the long answer. The short answer: the fixed point is at long measurement times, not at short ones, thus the steeper the algorithm, the worse the uncertainty at short measurement times, thus the worse the algorithm.

Just in case the long answer: please note that, as mentioned before, the lowest uncertainty is limited by the system. Whether it is some form of entanglement, which has a steeper time-dependency curve, or just a single coherence-limited quantum system as used here, which has a less steep curve, this is the base. The entanglement is better and steeper, which is probably what the reviewer is referring to. However, here, we look at the effect of a dynamic-range algorithm. In a perfect algorithm, at long measurement times, the best the algorithm can do is reaching the lowest uncertainty defined by the system. This has a certain scaling, in our example $T_{\text{meas}}^{-0.5}$. Going to longer measurement times, this scaling continues, once again: it is constrained by the system. When moving to shorter measurement times, the best is still following this dependency. If for any reason, for example to increase the range, at some point the uncertainty will deviate from this line, creating a steeper line, the uncertainty becomes worse. If there are two algorithms starting to deviate at the same point, then the one that has the steeper dependency will worsen its uncertainty faster than the algorithm with a less steep dependency. Thus, the steeper algorithm is worse.

To improve the clarity in this matter, we have removed earlier references to this point (original manuscript page 5 line 18 and page 5 line 60) and confined this explanation to the discussion (page 7 line 16) with an additional sentence. For further information, the reader is referred at the same place to SI 3.

Reviewer's comments: 9. *In Figure 3, the optimisation is for what? It appears the optimised curve has uncertainty worse than the A_0 curve (Fig. 3b).*

Our reply: Optimisation of the uncertainty. The optimised curve will always be worse than the curve for A_0 , which is the curve for the lowest uncertainty, as explained in the base-algorithm description (page 2 line 82). For the A_0 measurement, all measurement time is spent on this lowest uncertainty area, giving thus the lowest uncertainty. For any sequence spending any time, no matter how short (but more than 0 s), on any other sequence that has a higher uncertainty, the combined uncertainty will always be worse.

We have changed the wording of the original explanation (page 2 line 82), and we have repeated a similar explanation while discussing Fig. 2a, which lists the various important points (page 5 line 15). For the latter, we have split the original paragraph in two to focus in one paragraph on the various points. Moreover, we have adjusted the caption of Fig. 3b ("Minimised uncertainty etc") and we clarify what we optimise in the main text (page 5 line 37).

For completeness, one might wonder why not just use A_0 , since the optimised curve is simply worse. This is the topic of the paper, as described in the abstract/introduction (for example page 1 line 6, page 1 line 67 and page 2 line 55): the range of a measurement using A_0 only is minuscule. We seek to enlarge this range, while retaining the sensitivity as much as possible. We accomplish this with our algorithm: the range is large, while the uncertainty gets arbitrarily close (but once again will never surpass or be the same as) the lowest uncertainty possible given the system. For clarity, when discussing the first measurements, we have added a reminder (page 5 line 20). Moreover, we have clarified the caption of Fig. 3 by adding “large-range” and “small-range” to the various sequences.

To conclude, when comparing curves, it is important to compare curves with the same range. Thus the optimised curve should be compared with the single-area curve of the same range, which is the top grey one (Fig. 3b). The optimised curve is initially the same as the grey one, and eventually always below it. It is never above it. In comparison, a standard sequence such as in Fig. 2a (the combined green one) is not always below the grey curve.

Reviewer’s comments: 10. *In Fig. 3b, how could the relative difference be less than 1 at the long time limit (since in Fig. 2b the optimised uncertainty is almost one order of magnitude larger than the A_0 curve)?*

Our reply: There is no A_0 curve in Fig. 2b. Each line is described in the caption, but even without it, please note that there is no line in Fig. 2b that has the typical shape of any A_n line: flat and then decreasing. Maybe, the reviewer is confusing the Heisenberg limit curve being the A_0 curve. It is not: the Heisenberg limit assumes infinite T_2 , and thus is of course better than our measurements, since our T_2 is not that long yet.

To help to reduce the confusion, we have changed all colours in Fig. 2b to be the same as the ones in the SI that describes them (SI Fig. 3). Moreover, we have added a legend.

Reviewer’s comments: 11. *In the figures (both in the main text and in the SI), too many curves and symbols are plot in similar colours, similar shaped, overlapped, crowded, etc, very difficult to be distinguished by naked eyes.*

Our reply: Since no specific examples have been given, we start with a general reply. Logically, simulations and measurements have similar shapes, we cannot change any results just to make a figure look nicer. Moreover, when data and/or curves are put in the same figure, they are meant to be compared, and placing them in different figures would defy the meaning. For all curves that overlap or are on top of each other, that is generally the point: once again, everywhere the values relative to each other are rather important (so offsetting is not an option, except for Fig. 1d).

Please note that we chose the colours carefully. We are completely consistent with the usage, for example, anything related to area A_0 is always blue: the crosses in Fig. 1a, the area in Fig. 1b, the indicator in Fig. 1c, the distribution in Fig. 1f, the data and simulation result in Fig. 2a, the relative number of iterations in Fig. 3a, the plot for A_0 in Fig. 3b, the relative number of iterations in Fig. 3c, and the same for SI Fig. 7 and 8. In plots that are not related to A_0 , blue has been reused for something else, since the number of clearly distinguishable colours is limited. We do think that the choice for the symbols (a cross, a circle, a triangle, and a diamond) are fairly different.

Next, we discuss each figure, and the improvements we made.

Main text figure 1: we believe there are not many curves, all colours are consistent throughout the figure, and the only lines on top of each other in Fig. 1f are meant to be like that, and we think they are clearly visible. We have changed the second added line to dashed, and the third to dotted, so that at the overlapped places, they are all still visible. Also, we have changed the location of the label for A₀/2. Finally, we updated the symbols in 1c and 1d to be consistent with the four chosen symbols.

Main text figure 2: 2b and 2c are mostly whitespace, there are few lines, and all are meant to be compared. Fig 2b was updated as mentioned before (changed colours and symbols to the same as the referred SI 3), and consequently other symbols and colours are used for the data and simulation results. Also, a legend was added in the white space.

In Fig. 2c, the magenta line and dashed black line are basically on top of each other, which again is the whole point, but since the black one is dashed, they are both visible, and their relative position has been added to the caption. Moreover, for comment 6 of the reviewer, we have added the overhead data, which since there is barely any difference, is overlapping the non-overhead results. We made them grey and plotted them in the background to limit clutter.

2a is indeed a concern. The colours are consistent, and the lines are rather predictable. The combined sequence is much less clear. We have improved it by lowering the range of the axes, by removing the dark green data (since it is additional information relevant for specialists only), and by skipping half of the data points in the steep region of the combined sequence. This limits the overlapping data to the combined sequence, which is as expected and important to show, but we hope that it is much clearer now.

Main text figure 3: It is rather empty. 3a has five examples, each label is placed in isolation. The curves naturally overlap and cross each other, which is the point of this graph. Some additional labels are in black, and all are far from each other. 3b is very straightforward: 4 lines for the areas which do not overlap and use the consistent colour scheme. One optimum, which line naturally overlaps with the others, which again is the important point of the figure, but it is placed behind all lines so everything is visible. Other labels in black, far away from the rest. 3c: all labels far apart, and the small overlap for long measurement times does not obscure that the smallest areas' data points become fairly constant. We have changed the pluses to triangles and the dots to diamonds for consistency. 3d: relatively empty, just a few labels, and a gradient of lines. We have inverted the gradient, since the bottom line is more similar to A₀, improving the consistency with the colour blue.

SI figure 1: Basically single data plots with few annotations, we made no changes.

SI figure 2: Many lines are on top of each other indeed, which is exactly the point. If we would offset them, the whole point is lost. The similar colours for similar situations are on purpose, and they are differentiated with the line style. This is to group the number of phases via colour (one red, two blueish, four purpleish, eight black), while other differences, which are small, are indicated differently just to show that these differences are small. The caption also guides the reader step-by-step through the build-up of this figure (using descriptions as “upper” and “on top of”). Also, the three zoom levels were chosen to improve clarity. We made no changes to this figure.

SI figure 3/4: Very white-spaced figures. The choices for red/orange and blue/cyan are again simply because they are initially the same (which is straightforward from their description), and do not need to be distinguished. When they are different, this is clear, and the different symbols make this clear as well. We changed the colours though to be able to be consistent with the main text, and the symbols to have larger symbol-contrast.

SI figure 5: 5a and 5b are single data plots. Fig. 5c has four lines almost on top of each other except at one point, which is exactly the point of the figure, as described in the caption. We updated the caption to improve the message.

SI figure 6: New figure, but fairly consistent colour usage.

SI figure 7: As main text Fig. 3b/c. We have updated the symbols for consistency in 7a and 7c and relocated the labels for clarity.

SI figure 8: Similar to main text Fig. 3c, but simpler. We have updated the symbols for consistency.

SI figure 9: Single line figure with few annotations. We only shifted the leftmost label to increase the readability of the minus sign.

SI figure 10: New figure, single line with coloured areas. A zoomed-in version shows details.

SI figure 11: Single data figure with three insets in the giant white-space, we made no changes.

Reviewer's comments: 12. Page 2, line 81: Why is $T_{\{meas,M\}} = [1 + \log_2(M)] T_{\{meas,0\}}$? If $M=1$, would $T_{\{meas,1\}} = T_{\{meas,0\}}$? Again, what is the definition of $T_{\{meas,M\}}$?

Our reply: The situation is described using halved areas only. Thus if reducing with say 20, then 6 areas are used: 2^0 , 2^{-1} , 2^{-2} , 2^{-3} , 2^{-4} , and 2^{-5} (the latter gives 32 times, but 16 is not enough). This means we have to measure 6 sequences, and since as mentioned no optimisations are used for this example (so this gives what is included in $T_{\{meas\}}$), each needs their own period of the field. Thus the measurement time increases 6 times. This can be computed simply using the log2 function and ceiling the result, since once again, halved areas are used for this example: $\text{ceil}(1 + \log_2(20)) = \text{ceil}(5.3...) = 6$. The "1" is for the standard area, the " $\log_2(M)$ " for the additionally required areas.

Thus yes, if $M=1$, $\log_2(1)=0$, thus $T_{\{meas,1\}} = T_{\{meas,0\}}$. "M" is defined earlier as how much the area is reduced. If the area is not reduced, thus reduced with factor 1, the measurement time is indeed the same. $T_{\{meas\}}$ is defined before, M is defined before, so $T_{\{meas, M\}}$ is $T_{\{meas\}}$ for M. From a function point of view, we could write $T_{\{meas\}}(M)$, but M is not really a parameter that will be swept, but more of a choice, and hence we prefer to label $T_{\{meas\}}$ with M.

We have added a short explanation for the log2 (page 2 line 88), and a description before $T_{\{meas,M\}}$ (page 2, line 91). (We did the same for $\text{grad}_{\{max,M\}}$ at page 2 line 76.) Moreover, we have changed "0" to "1", since "0" meaning "the basic one which is the same as for $M=1$ " is not clear.

An additional note, given the reviewer's comments 3-4-6: these initial examples are for unfamiliar readers to ease them into the explanation of the algorithm. Details about overhead time are not important at this stage. Please note that when actually comparing with the standard method (the one described at page 2 line 70), we compare them fairly with the same overhead constraints.

Reviewer's comments: 13. The experimental method is not described.

Our reply: The main experimental method is using a confocal microscope and applying Hahn-echo like sequences. These are very well-known and described in depth in literature much better than we could, to which we also refer. Please do compare with other recent papers on the topic, only the new relevant parts/choices are described. For example in Ref. 36, they added real-time processing, which thus was described in the methods. They also describe their sample in the methods, which we describe in the main text (page 4 line 19). Moreover, anything non-standard, they use for example a solid-immersion lens, they mention in the methods, but we do not use anything special (they do not describe the standard confocal microscope or FID measurements). This is common practise, as NV centre measurements are a fairly mature technology nowadays.

From the algorithmic point of view, the measurements of Fig. 2a are well described (their sequence is given in the figure and in the paragraph at page 4 line 31). For the measurement of Fig. 2c we now explicitly mention the build-up of the sequence (page 6 line 14). The measurements of Fig. 2b are rather unclear and most complicated, and this is probably the best place to discuss how we optimised the sequences. Therefore, we have added another SI (SI 5) with the measurement details, which illustrates again the working of the algorithm (that more and more areas are added). Here, we also mention how we optimised the uncertainty (we used a simple brute-force approach). Including the points mentioned in reviewer's comments 6 and 7, this gives all information required by a different researcher to reproduce or reanalyse the experiments (experimental system, algorithm sequences, and definitions).

Reviewer #3

Reviewer's comments: *The authors devise a non-adaptive algorithm for high dynamic-range measurement of modulo-limited sensors. Principally, it can realize arbitrary dynamic-range measurement with retaining the highest sensitivity, which is practical for realistic applications. The algorithm is elaborated and the corresponding experiments and simulation are well done. However, the core idea of the algorithm does not have much novelty and has been used in common experiments by intuition. It can be illustrated simply: when the most sensitive way using coherence time adequately measures a large magnetic field beyond its dynamic range, it will lead to the ambiguity of multi-peak as shown in Fig. 1f; but it can be simply remedied by adding the experiments of using a shorter interference time but having a larger dynamic range to kick out wrong peaks; the process of kicking out is effective because it can use something like dichotomy resulting in a logarithmic time overhead; moreover, these remedying experiments do not require high precision, thus their duty cycle can be small enough at infinite measurement time. Since the journal of Nature Communication requires papers with enough novelty, I do not recommend this paper.*

Our reply: We would like to thank the reviewer for his/her time to read and comment on our manuscript. Moreover, we would like to thank the reviewer for the compliment that the manuscript was so well-written, that after reading it, the reviewer felt like our algorithm was so intuitive, that he/she already imagined common experiments utilising it. We regret the reviewer's conclusion though and we do think it is unfair, as we will elaborate on below.

Firstly, aren't most experiments based on a hypothesis, or in non-scientific terms potentially on someone's intuition?

Secondly, there are several nice publications about high dynamic-range algorithms. For example Refs. 34-36, all published in Nature Nanotechnology, which all implement an algorithm based of a previous paper (Ref. 30), which in turn was based on a previous paper (Ref. 31). All of these earlier papers (so also Ref. 37 and 38) do exactly the opposite of what the reviewer deems intuitive: more iterations of the short subsequences (in our case small areas), less of the long subsequences (in our case large areas). In other related topics, for example Ref. 33 in Ref. 38, the focus is also on the most-significant bits (with error-checking used on the most significant bits (equivalent to small areas) and using fewer measurements on the least significant bits). We, and many researchers in this field, find these papers all interesting and important. Since not everybody seems to share the reviewer's intuition (a broad audience including reviewer #2 needs many clarifications as well), the reviewer might agree that it is important to share our results and his/her intuition by publishing our manuscript, especially if it can be used in common experiments as well, instead of in quantum measurements only. Worthiness of publishing should be based on a broad audience, and not on a single person who happens to be intuitive towards the subject.

Thirdly, even for the few who have the right intuition about one aspect of the algorithm, this does not mean all aspects are easily guessed. How about the behaviour at short measurement time? The relative distribution of times amongst the smaller areas at long measurement time? How much time to spend on the largest area at long measurement times? The number of areas to use? The effect of the number of areas used? Which areas to choose? And intuition says to use "something like dichotomy resulting in logarithmic time overhead", but when decreasing measurement time for small areas, the opposing options are not that opposing anymore, how to deal with that (wrong peaks are actually not kicked out by default, see Supplementary Note 9)? If Higgs' intuition dictates Higgs boson exist, why should we measure it? Could intuition be wrong?

Finally, we like to share our thoughts on why to publish in this journal. Previous experimental realisations of existing algorithms were published in Nature Nanotechnology (Ref. 34 and 35), just as

an implementation of an adaptive version (Ref. 36). We devised an algorithm with the novelties to retain the highest sensitivity, and to increase the range beyond what is possible with a standard measurement, both unique compared to earlier approaches. Moreover, our approach limits the overhead time, also different compared to the previous best algorithm (we added this now in Fig. 2c and in the discussion page 7 line 83; the previous best Ref. 36 worsens its sensitivity almost by an order of magnitude when including overhead). Also, we explain the first pure Heisenberg-like scaling for such algorithms (see reviewer #1 page 1 middle paragraph starting with “A second important feature” for details). We describe the algorithm in great detail such that it can be used in other experiments (by using areas for description), and explore several parameters and explain them as much as possible with simulations (number of areas, specific areas, scaling), most of which were untouched by previous papers. Hence, we think that publication in Nature Communications is rather reasonable.

Below, we address the remainder of the comments of the reviewer.

Reviewer’s comments: *1. What combination of iteration numbers of each subsequence is used for blue crosses and red circles in Fig. 2b? It is different from the equally combined sequence in Fig. 2a.*

Our reply: The combination of iteration numbers in Fig. 2b is as given by our algorithm, thus optimising the uncertainty. This is different from an equally combined sequence such as in Fig. 2a, since this is merely a simple example suitable for explanation. We have added “Example” to the caption of Fig. 2a and “Optimised” to the caption of Fig. 2b to improve clarity.

To elaborate on the measurement of Fig. 2b, we have added Supplementary Note 5, to which we refer (page 5 line 89). Please note that these iteration numbers depend on the choice of range and the parameters of the used quantum system, thus they are only true for this specific experiment. For each experiment, these need to be calculated. Of course, given a specific sensor, these values only have to be calculated once. These details are mentioned in the new Supplementary as well.

Reviewer’s comments: *2. Why the uncertainties of the measurements with a single area are below the combined one as shown in Fig. 2a and Fig. 3b? Notice that the choice of a single area is also a special combination. I guess that the initial set uncertainties are different and within dynamic ranges respectively. It may cause confusion and need some explanation.*

Our reply: As mentioned in the text (page 3 line 25) and explained in Supplementary Note 1, since measurements of this nature have a limited range, there is a maximum uncertainty. This maximum uncertainty is when no measurement is done, so the probability distribution is flat, which given the constrained range computes to be $B_{\text{range}} / \sqrt{12}$. So even without measurement, this is the uncertainty. Therefore, at zero measurement time, the larger the range of the measurement, the larger the uncertainty. When increasing the measurement time, first, the sequence including the smallest area (for example one with only the smallest area, or our algorithm) will be repeated often enough to collect sufficient data (here photons) to start to give an estimate of the field. Hence, these will start to decrease first, while larger areas follow. Only eventually, for our algorithm, thus the optimum case, most time is spent on the largest area and thus its uncertainty gets arbitrarily close to this ultimate uncertainty, but do note that it will remain, even so slightly, above this ultimate uncertainty (Fig. 3b and the green line in Fig. 3c). For any non-optimal combination, such as in Fig. 2a, since it does contain non-largest area subsequences, it can never reach the same uncertainty as a sequence fully dedicated to this largest area. Please do remember that even while, for example in Fig. 3b, our algorithm (the optimum red line) and the largest area standard measurement (the blue

line) practically overlap for sufficiently long measurement time, only the former has a large range, while the latter has a small one.

This is already mentioned in the base-algorithm description at page 2 line 82, but we rewrote it to make it clearer. Moreover, since this is an important point, we have repeated the explanation in the context of the first measurements on page 5 line 15, and in addition we remind the reader about the point of the combined sequence, and that the range is proportional to the maximum uncertainty (page 5 line 22). Finally, we have update the caption of Fig. 3b/c to remind specifically which sequences have a large range, and which have a small one.

So only sequences with the same range should be compared. This would be the optimal one in Fig. 3b (red line) and the single-area one with the same range (top grey one). As can be seen, the single area one is at best the same as the optimal one (for short measurement times), but generally worse. One advantage of our algorithm is, as described in the reviewer's introduction, that for long measurement times it approaches the uncertainty of the most sensitive single-area sequence.

Reviewer's comments: *3. In the row 38, is the formula $B_{range}=1/(2B_{period})$ typed incorrectly? Should it be $B_{range}=B_{period}/2$?*

Our reply: We thank the reviewer for noting this mistake. We have corrected it (page 2 line 44).

List of revisions

In the manuscript.

1. Changed the title.
2. Page 1 line 5: DR -> range.
3. Page 1 line 9-10: changed wording.
4. Page 1 line 26: changed wording.
5. Page 1 line 37-38: DR -> range and changed wording.
6. Page 1 line 41-44: DR definition.
7. Page 1 line 64: DR -> range.
8. Page 1 line 68: DR -> range.
9. Page 1 line 68-73: rewritten for clarity, and relocated from the end of the paragraph.
10. Page 1 line 73-74: clarified what is measured.
11. Page 1 line 92: DR -> range.
12. Page 1 line 93: changed wording.
13. Page 2 line 5-6: adjusted for dynamic range definition.
14. Page 2 line 7-12: introduction new previous research (Ref 37).
15. Page 2 line 14: DR -> range.
16. Page 2 line 18: adjusted for dynamic range definition.
17. Page 2 line 24-25: added non-adaptive.
18. Page 2 line 25: DR -> range.
19. Page 2 line 26: addition for DR.
20. Page 2 line 27: changed wording.
21. Page 2 line 34: DR -> range.
22. Page 2 line 40: changed wording.
23. Page 2 line 44: corrected formula.
24. Page 2 line 56: added that a standard Hahn-echo is applied.
25. Page 2 line 68-70: rewritten.
26. Page 2 line 72: reminder for large-range.
27. Page 2 line 76: definition grad added.
28. Page 2 line 77: 0 label to 1.
29. Page 2 line 78: DR -> range.
30. Page 2 line 79: grammar.
31. Page 2 line 82-87: rewritten for clarity.
32. Page 2 line 88: added explanation log2.
33. Page 2 line 90-91: definition Tmeas,M added.
34. Page 2 line 92: 0 label to 1.
35. Page 2 line 99: symbols updated.
36. Page 2 line 105: panel reference updated.
37. Page 3 Figure 1:
 - a. Updated symbols in c and d.
 - b. Moved label in f.
 - c. Changed dotted->dashed and dashed->dotted in f.
 - d. Caption line 2: wording.
 - e. Caption line 8: wording.
 - f. Caption line 18: updated dashed/dotted.
38. Page 3 line 15: DR -> range.
39. Page 3 line 19-20: changed wording.
40. Page 3 line 22: DR -> range.
41. Page 3 line 30: DR -> range.
42. Page 3 line 32: removed initial reference to supplementary.
43. Page 4 line 2: removed reference to supplementary.

44. Page 4 line 4: added explicitly the phase shift.
45. Page 4 line 4-5: changed to “more homogeneous”.
46. Page 4 line 6-8: added explanation and reference for phases.
47. Page 4 line 8-10: rewritten given the inserted explanation.
48. Page 4 line 10-12: new reference to the supplementary and to new Ref. 38.
49. Page 4 line 26: changed wording.
50. Page 4 line 40-41: added reference to supplementary for shot-noise details.
51. Page 4 line 42: clearer mentioning overhead time.
52. Page 4 Figure 2:
 - a. Changed horizontal axis in a.
 - b. Limited the data points in a to reduce clutter.
 - c. Changed the colours and symbols in b to be consistent with the relevant supplementary.
 - d. Added a legend in b.
 - e. Added sensitivity with overhead in c in the background.
 - f. Caption line 1: added “Example” to clarify the reason for Fig. a, the symbol T_{meas} , and the frequency.
 - g. Caption line 4: made deviation plural.
 - h. Caption line 5: added “Optimised” to clarify the reason for Fig. b.
 - i. Caption line 6: added frequency and changed to “steep region” for consistency.
 - j. Caption line 6-8: updated colours and symbols.
 - k. Caption line 8: added infinite T_2 .
 - l. Caption line 10: changed wording.
 - m. Caption line 11: added frequency.
 - n. Caption line 12: added overhead information.
 - o. Caption line 12-13: added explanation newly plotted points.
 - p. Caption line 14: added overhead information.
 - q. Caption line last: added relative location of lines.
53. Page 5 line 1-2: indicated that in this case a single period is used.
54. Page 5 line 6-7: split paragraph in two.
55. Page 5 line 8-9: changed wording.
56. Page 5 line 15-23: added additional explanation of differences between sequences, and added reminder for which sequences have a large range.
57. Page 5 line 36: removed explanation algorithmic scaling.
58. Page 5 line 38-39: added what is minimised.
59. Page 5 line 42: changed wording for consistency.
60. Page 5 line 63: changed wording.
61. Page 5 line 64: changed to present tense.
62. Page 5 line 77: removed explanation algorithmic scaling.
63. Page 5 line 88-89: added reference to new supplementary.
64. Page 5 line 91: added infinite T_2 .
65. Page 5 line 91-92: DR -> range.
66. Page 5 line 92-94: added overhead time explanation.
67. Page 5 line 102: DR -> range.
68. Page 5 line 104: DR -> range.
69. Page 5 line 107: DR -> range.
70. Page 6 Figure 3:
 - a. DR -> range in b.
 - b. Updated symbols in c.
 - c. Inverted gradient of lines in d, such that blue is used for the line most resembling the largest area.
 - d. Caption line 2: areas A instead of A_n , since A_n is not really used.

- e. Caption line 3: clarified what is minimised.
 - f. Caption line 5: DR -> range.
 - g. Caption line 6-7: updated symbols.
 - h. Caption line 9-10: added small-range and large-range for clarity.
 - i. Caption line 12: updated colours.
 - j. Caption line 12-13: removed the explanation for the lower part to improve clarity, since it is given in the main text already.
 - k. Caption line 13: changed wording.
71. Page 6 line 2: DR -> range.
 72. Page 6 line 4-10: updated to make the point of increasing the range clearer.
 73. Page 6 line 13: DR -> range.
 74. Page 6 line 13-17: added the sequence information.
 75. Page 6 line 17-21: rewritten to add the overhead time information.
 76. Page 6 line 31: DR -> range.
 77. Page 6 line 32: removed the information about the areas, which is now given earlier.
 78. Page 7 line 16-19: now, only here the information about steepness of algorithms is given, and it is rewritten for clarity.
 79. Page 7 line 26-27: changed wording.
 80. Page 7 line 32-35: added visualisation of the negligible overhead.
 81. Page 7 line 61-62: updated for dynamic range meaning.
 82. Page 7 line 63-64: clarified the comparison.
 83. Page 7 line 73: changed rare stringed to strung.
 84. Page 7 line 78: added reminder for the reason to ignore stringing times.
 85. Page 7 line 83-87: added explanation for why stringing times can be ignored.
 86. Page 7 line 88-103: added paragraph to discuss frequency relevance.
 87. Page 8 line 5-6: changed wording.
 88. Page 8 line 9-10: changed wording.
 89. Page 8 line 13-15: changed explanation for clarify.
 90. Page 8 line 16: proper use of dynamic range.
 91. Page 8 line 28-29: DR -> range.
 92. Page 9 line 14, 17, 21, 24, 57: reference number changed.
 93. Page 9 line 51-56: new references.

In the supplementary.

1. Page 1: title and subtitles.
2. Page 2 line 7: added figure reference for explanation phases.
3. Page 2 line 8: repeated B.
4. Page 2 line 12: changed wording.
5. Page 2 equation 4: added the symbol for maximum uncertainty.
6. Page 2 Figure 1: changed wording in first line.
7. Page 3 line 7: added which line.
8. Page 3 line 7-9: improved clarity for what S is.
9. Page 3 line 25: DR -> range.
10. Page 3 line 30: added missing "supplementary" label.
11. Page 4 line 1-3: added shot-noise formula with reference.
12. Page 5 line 9-10: dynamic range corrected.
13. Page 5 line 19-21: new reference introduction.
14. Page 6 Figure 3: updated the colours and symbols, also in the caption.
15. Page 6 Figure 3 caption line 1, 13: changed wording.
16. Page 6 Figure 3 caption line 6: updated to using areas for consistency.
17. Page 6 line 12, 19, 20, 21: updated colours/symbols.
18. Page 6 line 16: changed wording.

19. Page 6 line 18: changed wording.
20. Page 7 line 3: changed wording.
21. Page 7 line 4, 7, 10, 16, 21, 23: updated colours/symbols.
22. Page 7 line 18-19: changed wording.
23. Page 7 line 17, 19, 23, 35: added small-range and large range labels.
24. Page 7 Figure 4: updated colours/symbols.
25. Page 8 line 3: added what it affects.
26. Page 8 line 5: DR -> range.
27. Page 8 line 11: wording.
28. Page 8 line 11-12: DR -> range.
29. Page 8 Figure 5 caption line 9: added relative description for lines.
30. Page 9: new supplementary with new figure for details of measurement of Fig. 2b.
31. Page 10 line 1, 3, 5, 8, 10, 12, 16: DR -> range.
32. Page 10 line 3: changed wording.
33. Page 10 line 8-10: updated to clarify the algorithm part and the experiment part.
34. Page 10 line 16-17: added timed pulses.
35. Page 11 Figure 7: Updated symbols for consistency.
36. Page 11 Figure 7 caption line 1, 2, 19: DR -> range.
37. Page 11 Figure 7 caption line 8: rewritten to clarify what is minimised.
38. Page 11 Figure 7 caption line 13: updated symbols.
39. Page 11 Figure 7 caption line 17: added small-range label.
40. Page 12 line 6-8: rewritten for clarity.
41. Page 12 Figure 8: updated symbols for consistency both in the figure and in the caption.
42. Page 13 line 12: DR -> range.
43. Page 13 Figure 9: moved left-most label for clarity.
44. Page 13 Figure 9 caption line 7: changed to inflection point for consistency.
45. Page 13 Figure 10: new to explain the overhead time (laser and stringing).
46. Page 13 line 15-16: new to explain the new figure.
47. Page 14 line 1-12: new to explain the new figure.
48. Page 15 line 5: added "sub".
49. Page 15 Figure 11 caption line 2: changed wording.
50. Page 16 line 5-6: added reference.

Reviewers' Comments:

Reviewer #1:

Remarks to the Author:

Authors have demonstrated a novel non-adaptive phase estimation algorithm (NAPEA) on NV spin qubit for sensing AC magnetic fields that can achieve higher dynamic range than previously demonstrated NAPEA algorithms, while retaining the optimum sensitivity. Authors have also adequately responded to the previously raised referee comments. However, the title ("Practically and theoretically unlimited...") reads a little awkward. Having either "experimental" or "technical" constrains would still make it not so practical anyways. Therefore authors should reconsider on the title and perhaps call this "Ultra-high dynamic range quantum measurements retaining high sensitivity". Otherwise, the manuscript is improved, and therefore I would extend my recommendation for publication.

Reviewer #2:

Remarks to the Author:

The revised manuscript and SI are improved and more readable. The key points are better explained (though not yet very clear). I understand that the authors have demonstrated a method to improve (without a limit in principle) the dynamic range of measuring the amplitude of an AC field (with a known frequency), without sacrificing the sensitivity (keeping the T^{-2} scaling with the "measurement time"). The method is to use a set of Hahn echo measurement of different durations (hence different modulo field amplitudes) and an optimized allocation of repeated cycles (for maximized information extraction). With the measurement phase areas A_n 's decreasing exponentially in the sequence (area halved in each subsequent measurement, e.g.), the "measurement time" needed to achieve a target modulo amplitude (which amounts to the dynamic range) increases only logarithmically with the dynamic range, and therefore the sensitivity (as measured by the field uncertainty times the sqrt of the "measurement time") is nearly unaffected. The work demonstrated and improved an existing idea and applied it to NV center sensing. It is certainly of interest in metrology and in the field of diamond sensing. However, I feel it is more suitable for a more specialized journal instead of NCOMM, for two main reasons: (1) The idea is not very original; (2) the definition of "measurement time" is less meaningful in the context of NV center sensing.

The first point is evidenced by the numerous works cited in the manuscript.

Let me explain the second point.

The authors did not give an explicit formula for the measurement time T_{meas} . After reading the manuscript and SI several times, I deduce/guess the definition as follows (if their definition is essentially different from mine, I would be willing to reconsider my recommendation):

T_n : the duration of a single Hahn echo sequence ($\pi - \tau - \pi - \tau - \pi/2$) for the phase area A_n , for $n=0,1,\dots, N$. (e.g., T_0 is just the period of the AC field. For each A_n , the measurement is carried out for K_n shots.

The whole sequence then can be repeated by M times (M can be a very large number) to achieve a high sensitivity.

It seems to me that the measurement used in the paper is

$$T_{\text{meas}} = K_0 T_0 + K_1 T_1 + K_2 T_2 + \dots + K_M T_M$$

For this definition, if all these T_n are much shorter than the NV center spin coherence time, the

phase estimation is in the coherent regime, where the T^{-2}_{meas} scaling is well understood.

However, such a scaling is not very meaningful for the NV sensing of an AC field.

In the NV sensing, one needs to prepare the NV center in each shot of phase measurement. Then in each cycle (T_{AC}) of the AC field, only one shot of measurement can be done (or at most a few phase-shifted measurement done for one A_n in different segments of a period of the AC field). Then, the NV center has to wait for the next AC field period. The waiting time has not been included in the measurement time. If one considers the waiting time between two measurements (which is unavoidable for using a single NV center sensor), the measurement time would become $T_{\text{AC}}(K_0 + K_1 + \dots + K_M)$.

In principle, one could apply all the measurements for different A_n simultaneously in one AC field cycle (then the measurement time, without considering repetition, is just T_{AC} , the period of the AC field), and the advantage of the algorithm demonstrated by the authors would be that the number of phase estimators needed for the simultaneous measurement for different A_n 's increases only logarithmically with the dynamic range. Such a multi-estimator method, however, does not look feasible for NV sensing.

Furthermore, the T^{-2} scaling is misleading. It is so only when the measurement is in the coherent regime, or in the case that the whole sequence needs not to be repeated $N \gg 1$ times. If we consider the whole data acquisition time T_D , the scaling will be the standard $1/\sqrt{T_D}$ for large N . That is the scaling really relevant if one is trying to achieve high precision by repeated measurements. The Fisher information has to scale linearly with the number of measurements if they are independent. For example, if one needs 10 minutes of data acquisition time to reach a precision of 1 nanoTesla in NV sensing, one cannot reach 10 femtoTesla precision using 100 minutes of data acquisition time.

In summary, the authors need to clarify what is the definition of T_{meas} in their algorithms and the experiments - Is it the duration of a Hahn-echo sequence (all the summation of the durations of different Hahn echo sequences for different A_n), or is it the total data acquisition time?

Reviewer #3:

Remarks to the Author:

Based on the revised manuscript and the reply from the authors, I am not convinced that the paper meets the requirement for publication in Nature Communications in the current version. Before the publication, the authors should clearly address these two concerns:

1. I agree with reviewer #1's comment and think "unlimited range" is not appropriate, even with the word "theoretically". Besides the finite pulse-length problem that reviewer #1 mentioned, because the two $\pi/2$ pulses are set at the non-zero magnetic field point, as the amplitude of the AC field becomes larger, the off-resonance effect becomes a problem especially when the resonant microwave seems to be the only way to control the spins. As a result, the $\pi/2$ is never a $\pi/2$ pulse and the inaccurate $\pi/2$ will make the sensing result incorrect. This is not a technical problem but theoretically exists. Though this inaccurate $\pi/2$ pulse can be corrected by changing the microwave frequency to on resonance if the magnetic field is known at this proper time, for an unknown AC field, the calibration cannot be done in principle.
2. Following the first comment, the simulation in figure 2c is less reasonable. The claimed range of 10^8 nT and the dynamic range of 10^7 is achievable. The authors should update figure 2c under the consideration of the off-resonance effect.

Reviewer #1

Reviewer's comments: *Authors have demonstrated a novel non-adaptive phase estimation algorithm (NAPEA) on NV spin qubit for sensing AC magnetic fields that can achieve higher dynamic range than previously demonstrated NAPEA algorithms, while retaining the optimum sensitivity. Authors have also adequately responded to the previously raised referee comments.*

Our reply: We thank the reviewer again for reading the revised manuscript and we are content that the improvements made the content clearer.

Reviewer's comments: *However, the title ("Practically and theoretically unlimited...") reads a little awkward. Having either "experimental" or "technical" constrains would still make it not so practical anyways. Therefore authors should reconsider on the title and perhaps call this "Ultra-high dynamic range quantum measurements retaining high sensitivity". Otherwise, the manuscript is improved, and therefore I would extend my recommendation for publication.*

Our reply: We agree that it reads awkward, but since we expected new comments from reviewer #2 anyway, we tried to experiment with the title a bit to find a suitable one. Indeed, since technical constrains could limit specific practical implementations, only the theoretical remains, but technically even a standard measurement (even though with rather bad sensitivity) would have a theoretical infinite range, though it would be harder to practically come to a similar range as any algorithm. Therefore, we changed the title to "Ultra-high range quantum measurements retaining its sensitivity". Please note the change to "its", since using the word "high" twice in the title is slightly awkward as well, and "its" is more appropriate, since a sensor with a low sensitivity would retain its low sensitivity with the algorithm.

Reviewer #2

Reviewer's comments: *The revised manuscript and SI are improved and more readable. The key points are better explained (though not yet very clear). I understand that the authors have demonstrated a method to improve (without a limit in principle) the dynamic range of measuring the amplitude of an AC field (with a known frequency), without sacrificing the sensitivity (keeping the T^2 scaling with the "measurement time"). The method is to use a set of Hahn echo measurement of different durations (hence different modulo field amplitudes) and an optimized allocation of repeated cycles (for maximized information extraction). With the measurement phase areas A_n 's decreasing exponentially in the sequence (area halved in each subsequent measurement, e.g.), the "measurement time" needed to achieve a target modulo amplitude (which amounts to the dynamic range) increases only logarithmically with the dynamic range, and therefore the sensitivity (as measured by the field uncertainty times the sqrt of the "measurement time") is nearly unaffected.*

Our reply: We thank the reviewer for reading and commenting on our manuscript, and we are happy to see in the summary above that the reviewer understands most of the principles now. There seem to be two main exceptions. The first, the idea that it is limited to AC/Hahn-echo sequences (for which we made no changes, as it is already mentioned at: page 2 line 32, page 5 line 74, the new paragraph added last time for this at page 7 line 95, and the conclusion is general). The second is the part with respect to scaling, but we will discuss that in detail at a specific comment about it below.

Reviewer's comments: *The work demonstrated and improved an existing idea and applied it to NV center sensing.*

Our reply: We think this is an unfair statement. The reviewer's statement sounds as if we found a paper with an algorithm (for example Ref. 30), then demonstrated it (for example as done in Ref. 34 and 35 for Ref. 30), and then improved it (for example Ref. 36). Of course, that would not be worth publishing again, since the mentioned excellent papers already did so. At best, if "increasing the range" is an existing "idea" (but it is more of an existing target), then the claim could be that this work had a new idea on reaching an existing target, and demonstrated this new idea.

Reviewer's comments: *It is certainly of interest in metrology and in the field of diamond sensing. However, I feel it is more suitable for a more specialized journal instead of NCOMM, for two main reasons: (1) The idea is not very original; (2) the definition of "measurement time" is less meaningful in the context of NV center sensing.*

Our reply: The main message of our manuscript is the introduction of a novel algorithm to tackle the range/sensitivity target for quantum sensors. As proof of concept, we use a single NV centre, but the algorithm is in no way limited to this. It is limited to modulo-limited sensors. The description we give is fully general, using areas for the description, and the experiments we perform on the NV centre are all standard, there are no NV centre-specific parts. Of course, for quantum sensing a quantum system is required that is readable, can be manipulated etc.

The two main reasons we will discuss below.

Reviewer's comments: *The first point is evidenced by the numerous works cited in the manuscript.*

Our reply: That numerous works are cited is evidence for the importance of the topic, however, claiming it is evidence for pure non-originality is debatable. Firstly, technically we suppose, if there is

even just one paper that mentions vaguely a related topic, any research about the topic is not very original. Please note that most papers are thus not very original in this sense of the word. Secondly, we think that a first paper on a certain topic, although interesting, is rarely the perfect paper about this topic with all the correct answers, and it doesn't mean that future papers about the same topic should already be considered mediocre at best. Papers should be judged on their content, not on when it was published. Just look at any journal, we should see how many pure original works are published on average. Pure original work is great, but given the advanced state of science nowadays, it is rather rare. Combined multidisciplinary efforts are essential in current scientific progress. As example, we can look specifically at our topic: the previously mentioned papers (Ref. 34, 35, 36) that the reviewer described as "work that demonstrated and/or improved an existing idea" (being Ref. 30) were all published in another excellent journal (Nature Nanotechnology). We do appreciate that in the opinion of the reviewer our work is more suitable for Nature Nanotechnology, but in our opinion the more general approach and the numerous advantages of our algorithm are rather suitable to be exposed to the broad audience of Nature Communications.

Reviewer's comments: *Let me explain the second point.*

Our reply: Please do, as we showed last time already that there is no practical difference between any measurement time that at least includes the phase-sensing time (which of course it should, since this is the essential part of the measurement). Moreover, since the reviewer already pointed out and used as an argument "numerous works cited in the manuscript", we would think that the reviewer is fully aware of the contents of these "numerous works", which fuels the confusion since, as also already explained last time, we follow suit with the definitions, also for measurement time. Nonetheless, we hope we can further clarify the confusion that apparently also exists on the reviewer's side. We will make some comments on various parts, and explain the overall details about measurement time and further clarifications we made to the text and supplementary afterwards at one place (at the "summary" of the reviewer), so please be patient. We do appreciate the lengthy explanation for a single question, as it helps us to understand where things go wrong.

To start with a very short answer to the summary question: when overhead time (this includes the waiting times of course) is included, in the steep region the relation becomes steeper (page 5 line 77), and in the standard region the sensitivity becomes a few percent worse (page 7 line 92).

Reviewer's comments: *The authors did not give an explicit formula for the measurement time T_{meas} .*

Our reply: Indeed, we wrote no explicit formula, as we use two logical descriptions, which are used when appropriate (and always as mentioned in the text): with and without overhead time. Please note again that this is common sense, please do look at the previous papers on this topic: all overhead time is always ignored in descriptions and conclusions, and overhead time is discussed separately. For example, in Ref. 35, they do give an explicit formula for the phase-sensing time, which is the measurement time excluding overhead time. This is all fine, as like we said last time, the properties of the algorithms should be investigated and compared, while overhead time is algorithm-independent and will decrease due to technical advances over time.

So the first of the two descriptions we use is the standard one: measurement time without overhead, thus the phase-sensing time. This is the measurement time that is relevant for sensing only, thus the time between the $\pi/2$ -pulses. This is important when discussing the properties of the algorithm, as overhead time (which depends on the system, implementation etc and is not algorithm-specific) would occlude the exact effect of the algorithm itself.

The second is the measurement time including all overhead time, since as mentioned before this depends on the system, this is for our specific example (so single NV centre). All means all: thus laser pulses, MW pulses, waiting times, idle times. Although the algorithm of course has some influence on how large the overhead would be (e. g. an adaptive algorithm would require computational overhead), these overhead times are essentially algorithm independent (e. g. if computers become incredible fast, the computational overhead time becomes negligible for every algorithm), and thus are only relevant for current implementations. Therefore, most overhead-related information is very specific and will be outdated within months (hopefully). Nonetheless, it is important to show a proof of concept and to give an understanding of the effect of overhead, hence we also show this information, but it is secondary to the exploration of the algorithm.

Of course, for AC, overhead time also includes the idle times. Since this is important to understand, we have a supplementary note dedicated to it, which we discuss again later.

Reviewer's comments: *After reading the manuscript and SI several times, I deduce/guess the definition as follows (if their definition is essentially different from mine, I would be willing to reconsider my recommendation): T_n : the duration of a single Hahn echo sequence (π - τ - π - τ - $\pi/2$) for the phase area A_n , for $n=0,1,\dots, N$. (e.g., T_0 is just the period of the AC field. For each A_n , the measurement is carried out for K_n shots.*

Our reply: A detail: as mentioned last time and in the discussion, T_0 is related to the largest area, which is not necessarily related to the period.

Reviewer's comments: *The whole sequence then can be repeated by M times (M can be a very large number) to achieve a high sensitivity.*

Our reply: A detail: increasing M does not necessarily improve the sensitivity, but it improves the uncertainty. So technically you could have the same sensitivity for $M=1$ and $M=10000$, but the latter would be required for the desired uncertainty.

Reviewer's comments: *It seems to me that the measurement used in the paper is $T_{meas}=K_0T_0+K_1T_1+K_2T_2+\dots+K_MT_M$*

Our reply: So as mentioned above, this is the measurement time without overhead, or phase-sensing time, essential for exploring the algorithm by itself, and always used in all previous papers on this topic.

Reviewer's comments: *For this definition, if all these T_n are much shorter than the NV center spin coherence time, the phase estimation is in the coherent regime, where the T^{-2}_{meas} scaling is well understood.*

Our reply: A detail: please note that the Heisenberg-like scaling is indeed well understood when simply elongating the delay between the $\pi/2$ -pulses in a single sequence in the coherent regime. Also, please note the mentioning of "Heisenberg-like" scaling, as, when T_2 would be significantly longer than the period of an AC field, when the delay starts to span multiple periods, the scaling is T^{-1} instead of T^{-2} , which is still Heisenberg-like scaling (it depends on the area vs delay relationship, see Fig. 1c). The exact power/shape depends on the shape of the field, but the scaling in the coherent regime is always called "Heisenberg-like scaling". Lastly, the scaling of combining multiple $\pi/2$ -delays is not that well understood, which is evident from the reviewer's comments, and to which we will come back later.

Reviewer's comments: *However, such a scaling is not very meaningful for the NV sensing of an AC field. In the NV sensing, one needs to prepare the NV center in each shot of phase measurement. Then in each cycle (T_{AC}) of the AC field, only one shot of measurement can be done (or at most a few phase-shifted measurements done for one A_n in different segments of a period of the AC field). Then, the NV center has to wait for the next AC field period. The waiting time has not been included in the measurement time. If one considers the waiting time between two measurements (which is unavoidable for using a single NV center sensor), the measurement time would become $T_{AC}(K_0+K_1+ \dots +K_M)$.*

Our reply: The reviewer's proposed measurement time would be the measurement time including overhead time for a straightforward, but not very meaningful, implementation. This is not recommended to use in actual implementations, and is discussed in detail in Supplementary Note 8 (to which we will get back below, since we updated it significantly as apparently it is not intuitive that few periods are required and that the overhead time has little effect on the sensitivity).

Reviewer's comments: *In principle, one could apply all the measurements for different A_n simultaneously in one AC field cycle (then the measurement time, without considering repetition, is just T_{AC} , the period of the AC field), and the advantage of the algorithm demonstrated by the authors would be that the number of phase estimators needed for the simultaneous measurement for different A_n 's increases only logarithmically with the dynamic range. Such a multi-estimator method, however, does not look feasible for NV sensing.*

Our reply: In principle, this is also an option. We do not explore this option, and many other options, though. We agree that probably for many applications, this might not be feasible. For example, as the different phase estimators are at different physical positions, maybe corrections are required, for which the origin of the field should be known. Anyway, we agree, but it is not relevant.

Reviewer's comments: *Furthermore, the T^{-2} scaling is misleading. It is so only when the measurement is in the coherent regime, or in the case that the whole sequence needs not to be repeated $N \gg 1$ times. If we consider the whole data acquisition time T_D , the scaling will be the standard $1/\sqrt{T_D}$ for large N . That is the scaling really relevant if one is trying to achieve high precision by repeated measurements. The Fisher information has to scale linearly with the number of measurements if they are independent. For example, if one needs 10 minutes of data acquisition time to reach a precision of 1 nanoTesla in NV sensing, one cannot reach 10 femtoTesla precision using 100 minutes of data acquisition time.*

Our reply: The scaling is mostly important for a conceptual point of view when combining multiple $\pi/2$ delays. As can be understood from Fig. 3a, it is possible to design any sequence with any time dependence during any span of measurement time. However, these will not be optimal, in general, and their uncertainty/sensitivity will be rather bad (for example for the T^{-4} scaling in Fig. 3a, or the T^{-5} scaling that is required for the reviewer's example). The importance of the discussed scaling is that for our algorithm, it is indeed the "coherent" scaling, and not scaling resulting from artefacts. This is discussed in the text (page 7 line 5). This scaling is true for the algorithm (while, even with infinite T_2 , the standard measurement will still scale as $T^{-0.5}$). On the other hand, for practical applications, when T_2 is not infinite, the reviewer is correct, the T^{-2} is not relevant. Please note though, that we are the first to report this: we already described this in the discussion (page 7 line 37). This affects any algorithm, as this is a limitation defined by the underlying system (thus: that the scaling is $T^{-0.5}$ for longer measurement times is not an effect of the algorithm, but of the limited T_2). Another way to look at this: if at any point one would forget about our algorithm and just repeat exactly the sequences measured so far, it will always scale as $T^{-0.5}$ (since it is merely a repetition of a measurement). The algorithm makes it scale at T^{-2} , but due to the limited T_2 the algorithm cannot continue indefinitely (scaling-wise).

Moreover, we do not think we are misleading in any way, just like previous papers on this topic. All graphs show clearly both regimes (Fig. 2a, 2b, 3a, 3b, 3d, Supp Fig. 3, 4, 5, 7b), and the descriptions mention it is for short times, and the supplementary goes into detail about the scaling and explains all parts (Supplementary Note 3). Also, as mentioned, we mention ourselves that we think that for practical applications, the scaling is not relevant (page 7 line 37). Moreover, we do not mention the scaling in the title, as although it is an in-principle interesting point, it is not a main point. The abstract has to be short and should shortly give all relevant topics, but of course the abstract is not the place to explain all details. We say the algorithm approaches the scaling (an algorithmic point), we do not say that the scaling also happens when limitations of the underlying system are taken into account (a standard point, which is not something to mention in an abstract), or that it is a major point of advantage: we just mention we will discuss it (please note again, the scaling is introduced in the abstracts of Ref. 34 and 35 in a similar fashion, which is again true for their algorithms, we do not think they are, or meant to be, misleading, as a reader should not jump to incorrect conclusions based on the concise abstract). We discuss the scaling of previous papers in a similar manner (Page 1 line 86, 91, 98, Page 2 line 4), and we mention it everywhere (page 5 line 16, 46, 58, 67; page 5 line 76 for the Supplementary Note reference; page 7 line 43).

Reviewer's comments: *In summary, the authors need to clarify what is the definition of T_{meas} in their algorithms and the experiments - Is it the duration of a Hahn-echo sequence (all the summation of the durations of different Hahn echo sequences for different A_n), or is it the total data acquisition time?*

Our reply: Here we would like to answer this concluding question, and explain what we have done to improve the clarity further. Simply put: all algorithm-exploring simulations/measurements only include the algorithm-independent times (thus $\pi/2$ -pulse to $\pi/2$ pulse), as is standard, and the practical measurement time is left for the discussion and the supplementary (so the total data acquisition time). Please note that this was already mentioned everywhere (but apparently "no overhead" and "all overhead" were not clear). Therefore, we changed the delivery of this information to remove all confusion. Moreover, the differences between the two measurement time definitions is small (for our algorithm) and not relevant for practical implementations.

Now, at page 5 line 5, we immediately state that overhead time is ignored at first, and that it will be discussed in the discussion. Since it is stated generally now, we removed a number of other places we mentioned it to improve clarity (Fig. 3d and its original reference at page 5 line 68). However, at the measurements we kept the information about the overhead times (so essentially repeating it), as we think it is important for non-familiar readers to understand why they are ignored (page 5 line 94, 97, page 6 line 23). All information in the discussion remains of course (page 7 line 16, 30-36, 60-61 about processing overhead, 66-94 about data-acquisition measurement times).

Moreover, we have significantly expanded Supplementary Note 8, which discusses overhead time in detail. Now, to make it visually completely clear, we plot the complete sequence required to measure the largest range in the new Supplementary Fig. 11, as we feel from the above explanation that the reviewer (and thus also others) cannot visualise how several areas would be combined. Please note that for smaller ranges, the sequence is almost always a subset. This is clear from another new Supplementary Fig. 10 that we added: we performed the measurements using the shown "compact sequences", and we plotted the results for all three defines of measurement times in this new graph. The three defines are:

- 1) Without overhead, thus the phase-sensing time, thus in the reviewer's terms: $T_{sequence} = K0T0 + K1T1 + \text{etc}$. Its sensitivity is computed using $T_{meas} = N \times T_{sequence}$.
- 2) With actual overhead, thus T_{meas} is the total data acquisition time. If one sequence takes M periods of the AC field, then it is $T_{meas} = N \times T_{sequence} = N \times M \times T_{AC}$.

3) With extreme overhead, useful for explanation, but not for actual implementations. Thus $T_{\text{sequence}} = T_{\text{AC}} \times \# \text{phase} \times \# \text{areas}$, with again $T_{\text{meas}} = N \times T_{\text{sequence}}$.

We updated the main text that references to this figure (page 7 line 86). Also, for completeness, we plot all the three different sensitivities in main text Fig. 2c as well (so we added number three, this new one we reference at page 7 line 74).

Thus one can conclude again, that for practical implementations, for our algorithm given the current state of technology, the sensitivity using the total data acquisition time is only a few percent (here 5%) worse than the ideal sensitivity that ignores all overhead. In case the reviewer is wondering why it changed from 3.8% to 5%: now, we use the new measurements for this comparison. While processing these measurements, firstly, we allowed the ideal sequence to always centre around the inflection point (as opposed to the actual measurement). Secondly, the largest area is slightly smaller than the full area of a sine to allow for the laser pulses etc to fit in the compact sequence (which is the sequence that is actually measured). Even though this ideal case is actually unrealistic (even with perfected technology), we only count the time from $\pi/2$ -pulse to $\pi/2$ -pulse for the ideal case, which is thus slightly shorter than last time. In other words, the ideal sequence is the hypothetically shortest sequence possible with the exact same areas as the compact sequence. Therefore, it gains another percent.

We hope the reviewer now understands that on the one hand, it is important to explore the algorithm without being affected by artefacts introduced by algorithm-independent overhead times. While on the other hand, for practical implementations as mentioned by the reviewer, the overhead time (which includes everything, also waiting times) barely worsens the sensitivity. Please note again that, as mentioned last time, even though it has little influence for our algorithm, overhead time is generally ignored in conclusions, as it is algorithm independent: all results of implemented algorithms would improve with new technologies. Once again, we would be able to claim a third order of magnitude of improvement over the previous best (Ref. 36, which has large overhead time, thus much longer data acquisition times), but we choose not to do so, as we think including algorithm-independent parts into the comparison is unfair. (And, for an older paper like Ref. 34, who also show a plot excluding and including overhead time, there is also an order of magnitude extra difference.)

In summary, the performance of our algorithm using total data acquisition time is practically the same compared to our algorithm without overhead time, and even more superior to other algorithms given the current technology, but we think it is completely irrelevant since this is ultimately algorithm-independent and this specific difference should and will decrease over time with technological advancement.

Just in case it is still not clear, in formula form:

$$\sum_{i=0}^M K_i T_i \approx \text{total data acquisition time},$$

and please do not forget:

$$\text{sensitivity} \propto \sqrt{\text{measurement time}}.$$

Reviewer #3:

Reviewer's comments: *Based on the revised manuscript and the reply from the authors, I am not convinced that the paper meets the requirement for publication in Nature Communications in the current version. Before the publication, the authors should clearly address these two concerns:*

Our reply: We thank the reviewer for reading our manuscript in depth and giving some useful comments, which we will discuss below.

Reviewer's comments: *1. I agree with reviewer #1's comment and think "unlimited range" is not appropriate, even with the word "theoretically". Besides the finite pulse-length problem that reviewer #1 mentioned, because the two $\pi/2$ pulses are set at the non-zero magnetic field point, as the amplitude of the AC field becomes larger, the off-resonance effect becomes a problem especially when the resonant microwave seems to be the only way to control the spins. As a result, the $\pi/2$ is never a $\pi/2$ pulse and the inaccurate $\pi/2$ will make the sensing result incorrect. This is not a technical problem but theoretically exists. Though this inaccurate $\pi/2$ pulse can be corrected by changing the microwave frequency to on resonance if the magnetic field is known at this proper time, for an unknown AC field, the calibration cannot be done in principle.*

Our reply: The reviewer makes a good point and there are several aspects to discuss.

Firstly, maybe the definitions of "technical" etc are user-dependent and not very clear, but what we mean is that it is algorithm-independent. The off-resonance pulses etc are a problem for an implementation of any algorithm (when it comes to high fields), but say if one could prepare/readout the spin differently (maybe), or if we can develop any engineering solutions (synchronised screening), the implementation of any algorithm would benefit.

Secondly, and we think a single line would suffice for the reviewer, but with respect to "the calibration cannot be done in principle": the core principle of a sensor is that it responds differently depending on the value of the quantity to measure. Just in case a longer explanation: any sensor can be modelled as a black box with the quantity to sense as input, and some output. Inside the black box, at least one component, but potentially several, have a change in response depending on the value of the measured quantity, changing the output of the black box. Now, to calibrate a sensor, it is placed in a situation where the quantity is known, and the output is recorded. This is done for the whole range of desired values, so the relation between input and output is known, and the sensor is calibrated. This is possible for any sensor (or even any black box), and whether it is a suitable sensor would depend on the determined relationship. In our case, of course ideally, there is only effect on the "component" spin during the time between the $\pi/2$ -pulses, which would make calibration and analysis very simple. However, if the exact working of the "component" microwave pulses also depends on the field, even though this makes the calibration and analysis more cumbersome, this still allows to calibrate the sensor, as there are just more "components" that respond to the field.

Thirdly, the details of the off-resonance effect in our measurements are discussed in the next point. (The change with respect to the title is discussed there too.)

Reviewer's comments: *2. Following the first comment, the simulation in figure 2c is less reasonable.*

Our reply: For starters, it is a measurement. We realise we did only indirectly mention this in the caption, thus we updated the caption. In the main text, it is clearly stated though (page 6 line 15).

Following the first comment, we would like to discuss it in more detail. The largest area that mostly defines the sensitivity is barely affected by the off-resonance problem, since the pulses are applied at the inflection points. This is for example visible in a simulation using the Hamiltonian of the spin for a Hahn-echo sequence under the exact circumstances. Even for high fields, there is barely any effect on the oscillation with magnetic field strength. We show the simulation results in Reviewer Fig. 1 at 0 T and 100 mT (simulating this whole range would be too time-consuming due to the high oscillation frequency). As a different way to make this intuitive, when looking at previous research where they used a DC field with a range of 16 mT (Ref. 35), they do not find to have a negative effect (except for an offset in the calibration). In our case, for a 2 kHz field, the effective field during the $\pi/2$ -pulse is about two orders of magnitude smaller even for a 100 mT field, so it is expected to have a rather small effect on the sensitivity, as Reviewer Fig. 1 confirms. However, for much larger fields, it would become a problem if no precautions are taken. On the other hand, we could look at the effect on the range. For the range defining subsequences, the $\pi/2$ -pulses are also still in relatively small fields (but not at the inflection points). As seen in Reviewer Fig. 2, for the measurement, it still barely effects it. But once again, this would worsen for larger fields. The main problem is the other large areas. They can have their $\pi/2$ -pulses at the maxima, which are the worst places to have them. Technically, one can measure the same area from inflection point to inflection point with a FID sequence, but we didn't do that here. The larger areas are affected (at high fields), but since their effect on the result is relatively small (they mostly contribute a bit to the sensitivity, and barely to the range), as can be seen in for example Supplementary Note 4 (and its figure Supplementary Fig. 5c: mostly in the steep region the difference is visible), the sensitivity will only worsen slightly (for example the error bar in Fig. 2c is about 10-20%, while the difference in sensitivity with less areas is less than a percent). It is still slightly visible, as a simulation would have the black dashed line and the magenta line in Fig. 2c exactly on top of each other (visually, given the log scale), while now there is a visible offset.

Reviewer Fig. 1: Very low/low/high field response of most-sensitive (largest area) part of sequence used for Fig. 2c. There is a single high-frequency oscillation which amplitude is effectively constant. The middle and right figures contain this high-frequency oscillation for a large range on the x-axis, making it look like a rectangle of data points, which thus only shows the amplitude.

Reviewer Fig. 2: Field response of largest-range (smallest area) of sequence used for Fig. 2c.

Since the focus of the manuscript is on the algorithm, most of the space is dedicated to explaining this. Discussing overhead time is already borderline, since it is implementation-dependent (like laser pulses) and algorithm-independent, but it is important for actual implementations (as for example adaptive algorithms would need processing power, so it is not completely algorithm-independent, but with “ideal” technology, it is). However, technical details of such implementation are outside the scope of what we want to present in this manuscript. We do want to make sure the reader does get a proper picture, even for implementations, so for this important point to consider while implementing this algorithm, we have added a reminder for the readers in the discussion (page 8, line 1).

Finally, the question remains whether this will ultimately limit the range for this algorithm. We still reckon other factors (probably the current state of technology) will limit it first. The off-resonance problem is namely sufficiently solvable (for this application). The details of such a solution with “broadband” pulses, a different topic, we would like to leave for a different, more technical, paper in a different journal (if it has not been published yet already by someone else, in which case we could just use their citation here). Nonetheless, we will give a simple example of the principle here. As this is fairly concise and not the topic of the current manuscript, we do not expect any comments or suggestions, but they are of course welcome.

On “broadband” pulsing

We will explain this with a specific example to limit the simulation time. Here, we choose to measure a DC field (since it has a stronger effect on the $\pi/2$ -pulses), and we use a very short time delay (because this decreases the frequency of the readout of the spin with respect to field magnitude, thus limiting the required data points in the simulation). Also, we use relatively long $\pi/2$ -pulses, to strengthen the off-resonance effect further, and we look at positive field magnitudes. Please note that none of these change the generality of the results. In Reviewer Fig. 3, the effect off-resonance has on this example measurement is visualised in a simulation result (using a Hamiltonian for the spin-1 of NV⁻): for low field it works as desired, while for high field the amplitude decreases. The latter is for two reasons. The first is that the Rabi contrast decreases since the axis of rotation “tilts”, thus the envelop decreases. Secondly, “revivals” appear in the amplitude, because the Rabi frequency increases in higher field, thus the “ $\pi/2$ ” pulse becomes inaccurate and ends in a different phase of this oscillation. It is clear that at high field, the contrast is basically gone, so sensing would not work. Now, there are several ways to combat this, and we present the basic principle here. Instead of applying just a single frequency, we could apply multiple (or multiple pulses, but we will focus on the former here, which is arguably less versatile, but simpler). A simulation for a simple example, multiple evenly spaced frequencies, is displayed in Reviewer Fig. 4. The range of fields for which the sensor would work has increased significantly. Please note that for this simple example the amplitude decreases. Moreover, although the analysis does have to take the shape into account, a sensor is actually fairly robust against small variations in the amplitude, since multiple phases (of the MW pulses) are used, and mostly the phase (of the oscillation) defines the result. The areas with MW-pulse phases that are at their maxima barely contribute to the result anyway (but would be affected by a change in amplitude), while the areas with MW-pulse phases that are at the linear parts do contribute (but the oscillation’s phase, and thus result, is not affected by a change in amplitude). More accurately, for range-defining measurements, small amplitude changes are fairly irrelevant, while for sensitivity-defining measurements, the sensitivity would become less homogeneous. When adding more frequencies, one should take “overlap” effects into account, but it is possible to create a large amplitude over a large range. (Alternatively, several sets of “non-overlapping” frequencies could be used, which lowers the sensitivity if more periods are required to measure the whole set.) Probably the most important point when using this for sensing purposes: one should for the calibration and analysis keep in mind that the phases (not MW phase, but the phase of the oscillation in Reviewer Fig. 4) can change over field magnitude (it is not just a single frequency with a fixed phase, as in Reviewer Fig. 1/2).

Reviewer Fig. 3: Field response of DC sensing for described example.

Reviewer Fig. 4: Field response of DC sensing for described example with multiple-frequency pulses.

Reviewer's comments: *The claimed range of 10^8 nT and the dynamic range of 10^7 is achievable.*

Our reply: We agree, which is why we showed it. Larger ranges will become more difficult without taking appropriate measures (but larger dynamic-ranges are probably possible with for example NV centre ensembles by improving the sensitivity for a given range). Another problem for example is that the NV centre is a spin-1 system, so when using it "as a spin-1/2 system", at some point the -1 and +1 transitions will "overlap" in different fields. Since for any system, without technical ingenuity it is hard to keep increasing the range, even with our algorithm, we updated the title.

Reviewer's comments: *The authors should update figure 2c under the consideration of the off-resonance effect.*

Our reply: This part was essentially discussed above (at the beginning of point 2).

List of revisions

In the manuscript.

1. Changed the title.
2. Page 4 Figure 2:
 - a. Caption of b near the end: removed “Please note that the measurements scale not as $T-2$ yet, since the limit is approached slowly”, since this is already described in the main text and not necessary in the caption, and the word count should remain below 350 words (for c and d below).
 - b. Added “full-period” overhead results in 2c.
 - c. Caption of c now mentions it is a measurement.
 - d. Caption of c now describes the additional data points in its graph.
3. Page 5 line 5: added general remark about ignoring overhead time at first.
4. Page 5 line 68: removed mentioning overhead since it is mentioned in the beginning.
5. Page 5 line 94: updated the reminder about overhead time.
6. Page 5 line 97: any -> all.
7. Page 6 Figure 3:
 - a. Caption of d: removed “In this simulation, the overhead time is ignored in order to focus on the effect of the algorithm”, since it is already mentioned in general, and we do not want to confuse the reader.
8. Page 6 line 23: added still in the reminder about overhead time.
9. Page 7 line 74: indicate the added sensitivities with impractical full-period measurement time.
10. Page 7 line 86: updated the information from renewed Supplementary Note 8.
11. Page 8 line 1: added reminder for readers about off-resonance pulses.
12. Page 8 line 25: added high for consistency with the title.
13. Page 8 line 27: updated conclusion to reduce confusion with respect to algorithm vs implementation.

In the supplementary.

1. Changed the title.
2. Page 13 line 9: removed “much”, since it is only slightly less efficient.
3. Page 13 Figure 9:
 - a. Added why the DC component cancels, near the end of the caption.
4. Page 14 line 4: changed overhead to time delays, since that is the relevant point here.
5. Page 14 line 6: added idle times to clarify that not only waiting times between pulses are meant.
6. Page 14 line 9: updated the reference to the new figure.
7. Page 14 Figure 10: new.
8. Page 14 line 11: new paragraph to explain the new figure (10).
9. Page 15 Figure 11: renewed version of the previous figure 10, now drawing the complete sequence for clarity.

Reviewers' Comments:

Reviewer #2:

Remarks to the Author:

The authors have clarified all the technical issues I raised and those the other reviewers raised. Since the issue of "originality" is quite subjective, which probably annoyed us all as authors, I am willing to concede. I can recommend the publication of the paper.

Optional suggestion of revisions:

If the length allows, it would be good to add the condition for the T^{-2} scaling in the abstract.

Reviewer #3:

Remarks to the Author:

In the reply, the authors propose a method to answer my concerns about the off-resonance effect. In this method, they use multi frequency microwave instead of the single frequency microwave to avoid the contrast reduction. Though this method is slightly complicated to implement in the experiment, it is a feasible solution to close the loophole. I think the revised version can be published.

Reviewer #2

Reviewer's comments: *The authors have clarified all the technical issues I raised and those the other reviewers raised. Since the issue of "originality" is quite subjective, which probably annoyed us all as authors, I am willing to concede. I can recommend the publication of the paper.*

Our reply: We thank the reviewer for reading the revised manuscript a final time. Since originality is subjective to some extent, we always compare with previously published papers to position a new manuscript in an as objectively as possible way. However, of course journals change their views over time, and anybody, including reviewers, could disagree with whether the previously published papers are positioned correctly, so this always remains a difficult task.

Reviewer's comments: *Optional suggestion of revisions: If the length allows, it would be good to add the condition for the T^{-2} scaling in the abstract.*

Our reply: We added the condition "in the coherent regime" to the abstract, and by removing the final line, the abstract remained within 150 words.

Reviewer #3

Reviewer's comments: *In the reply, the authors propose a method to answer my concerns about the off-resonance effect. In this method, they use multi frequency microwave instead of the single frequency microwave to avoid the contrast reduction. Though this method is slightly complicated to implement in the experiment, it is a feasible solution to close the loophole. I think the revised version can be published.*

Our reply: We thank the reviewer for the patience to read the revised manuscript a last time.